# COMMUNICATION-EFFICIENT MULTI-DEVICE INFERENCE ACCELERATION FOR TRANSFORMER MODELS

## ABSTRACT

Transformer models power many AI applications but suffer from high inference latency, limiting their use in real-time settings. Multi-device inference can reduce latency by parallelizing computation. Yet, existing methods require high inter-device bandwidth, making them impractical for bandwidth-constrained environments. We propose ASTRA, a communication-efficient framework that accelerates Transformer inference through a novel integration of sequence parallelism and a Mixed-Precision Attention mechanism designed to minimize inter-device communication. ASTRA compresses non-local token embeddings via vector quantization and preserves task accuracy through two optimizations, Noise-Augmented Quantization and Distributed Class Tokens. Experiments on ViT and GPT2 across vision and NLP tasks show that ASTRA achieves up to $2.64\times$ speedups over single-device inference and up to $15.25\times$ speedups over state-of-the-art multi-device inferences, while operating under bandwidths as low as 10 Mbps.

## 1 INTRODUCTION

Transformer models (Dosovitskiy et al., 2020; Devlin et al., 2019; Radford et al., 2019) have become central to modern AI applications in both vision and language domains. As models grow in scale to improve accuracy, their inference time becomes prohibitive, particularly for real-time applications or latency-sensitive user experiences. To address this bottleneck, many works have focused on optimizing single-device inference through techniques like quantization (Liu et al., 2021), pruning (Kwon et al., 2022), and knowledge distillation (Lin et al., 2022). Since the latency improvement on a single device remains fundamentally limited by the device's capacity, *Multi-device inference* draws increasing attention (Hu & Li, 2024; Du et al., 2024). It complements the above optimization techniques by parallelizing model execution across multiple devices. In theory, it can reduce latency nearly linearly with the number of devices. This setup is especially attractive in practical settings where resource-constrained edge devices or distributed consumer-grade hardware can collaborate to process sporadic inference requests.

Our analysis, however, reveals a critical bottleneck that has been overlooked: in bandwidth-constrained settings (<=100 Mbps), inter-device communication dominates the existing multi-device inference time, accounting for 58.6-93.5% of total latency (see Section 4.3). This communication overhead negates the theoretical speedups promised by existing multi-device approaches. Existing methods, such as Tensor Parallelism (TP) in Megatron-LM (Shoeybi et al., 2019), Sequence Parallelism (SP) in Voltage (Hu & Li, 2024), and Block Parallelism (BP) in DeTransformer (Du et al., 2024), rely on high-volume communication bandwidth to achieve modest speedups, as shown in Figure 1. This requirement far exceeds what is commonly

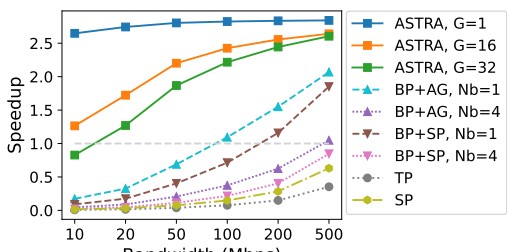

Figure 1: Latency speedup for existing multi-device inference methods and our proposed method ASTRA with different groups $G$ under different bandwidths with 4 devices and 1024 input tokens. BP: Block Parallelism, SP: Sequence Parallelism, TP: Tensor Parallelism, AG: AllGather. A smaller $N_b$ means less communication for BP.

available in bandwidth-constrained environments. For example, indoor Wi-Fi networks (e.g., Wi-Fi 4/5/6) typically deliver practical throughput in the range of 50–300 Mbps, depending on interference, distance from the router, and device capabilities. As a result, current approaches are ill-suited for wireless or edge deployments, where lower and more variable bandwidth is the norm (wii, 2025).

In this work, we propose ASTRA, a communication-efficient framework for accelerating Transformer inference across multiple devices. ASTRA rethinks how attention computation is distributed by building on sequence parallelism, which assigns different input tokens to different devices. The key innovation in ASTRA is a **Mixed-Precision Attention** mechanism that dramatically reduces communication cost: local attention is computed using full-precision embeddings, while remote tokens are encoded using low-bit vector quantization before transmission. These compressed embeddings are decoded on the receiving end and used for approximate attention computation.

To preserve accuracy, ASTRA incorporates two key designs. First, a **Noise-Augmented Vector Quantization** strategy injects Gaussian noise to vector-quantized token embeddings during training, enhancing the diversity of the quantized feature space and improving generalization to unseen inputs. Second, a **Distributed Class Token** scheme assigns each device its own local class token, which attends to uncompressed local tokens with full precision and vector-quantized tokens from other devices. The outputs from these distributed class tokens are then aggregated as a more comprehensive embedding before the final prediction. Through these techniques, ASTRA makes multi-device inference viable in low-bandwidth scenarios while maintaining high accuracy.

Our main contributions are:

- ASTRA— a new multi-device Transformer inference framework that significantly reduces communication overhead via a novel Mixed-Precision Attention mechanism. We design two techniques, Noise-Augmented Quantization and Distributed Class Tokens, to maintain high accuracy under aggressive communication compression.
- *Comprehensive Evaluation* — We evaluate ASTRA on Transformer models, including ViT (Dosovitskiy et al., 2020) and GPT2 (Radford et al., 2019), on both vision tasks (e.g., CIFAR-100 (Krizhevsky et al., 2009) and ImageNet-1K (Deng et al., 2009)) and NLP tasks (e.g., Wikipedia and Wikitext-103 (Merity et al., 2016)). ASTRA reduces the bandwidth requirement for multi-device speedup from 500 Mbps to as low as 10 Mbps, achieving up to $2.64\times$ end-to-end latency speedup even under constrained bandwidth where prior methods fail (Figure 1). It maintains task accuracy within $3.58\%$ of the original model under extreme communication compression. Moreover, ASTRA remains effective in heterogeneous device environments, achieving accuracy within $1.43\%$ drop under imbalanced token distributions across devices. It also retains compatibility with single-device quantization: when applied jointly, ASTRA achieves an additional $1.35 - 2.73\times$ latency speedup over single-device quantization alone, while preserving accuracy within $3.37\%$ drop.
- *Design Analysis* — Through detailed ablation studies, we demonstrate the effectiveness of each key design component. The Mixed-Precision Attention Computation Mechanism achieves a latency speedup of up to $171.82\times$ compared to the original attention under constrained bandwidth, the Noise-Augmented Vector Quantization improves accuracy by up to $0.86\%$ compared to naive vector quantization, and the Distributed Class Tokens design further enhances accuracy by up to $7.13\%$ compared to using a single class token.

## 2 BACKGROUND AND RELATED WORK

This section introduces existing techniques for multi-device inference acceleration and then the vector quantization (VQ) technique that is necessary to understanding ASTRA. A broader survey of related work is provided in Appendix A.

**Parallelization in Multi-Device Inference.** To accelerate inference latency, many recent approaches explore distributing Transformer computations across multiple devices. Techniques originally developed for training, such as data parallelism, pipeline parallelism, and model parallelism, have been repurposed for inference. *Data parallelism* (Li et al., 2014) and *pipeline parallelism* (Huang et al., 2019) improve throughput by processing different input samples or partitioning layers across devices. However, these methods are primarily suited to batch processing and offer limited benefits for latency-sensitive, single-request inference. *Model parallelism*, in contrast, splits the computation

of individual model layers across devices, enabling true per-request acceleration. Notable examples include: Megatron-LM (Shoeybi et al., 2019), which partitions model weights (tensor parallelism), Voltage (Hu & Li, 2024), which partitions token sequences (sequence parallelism), and DeTransformer (Du et al., 2024), which decomposes layers into smaller blocks for distributed execution. These approaches reduce some communication costs by tailoring the parallelization granularity, but they still require substantial bandwidth, such as 500 Mbps, to realize modest speedups. While this bandwidth may be achievable in high-performance server clusters, it exceeds the practical limits of many real-world environments, such as indoor Wi-Fi, where sustained throughput often falls lower than 300 Mbps depending on interference. Our work diverges from prior efforts by focusing explicitly on reducing the communication cost, which we identify as a primary bottleneck in multi-device Transformer inference.

**Vector Quantization (VQ).** ASTRA leverages vector quantization to compress token embeddings before inter-device transmission, significantly reducing communication overhead. VQ maps continuous vectors to discrete indices in a fixed codebook, allowing compact representation of information with a small number of bits (Gersho & Gray, 2012).

*Vanilla VQ* partitions the feature space into $K$ clusters, each represented by a centroid. Let $\mathbf{e} \in \mathbb{R}^{K \times D}$ be the codebook of centroids. For an input vector $\mathbf{z} \in \mathbb{R}^{D}$, VQ selects the closest centroid by $i = \arg\min_i \|\mathbf{e}_i - \mathbf{z}\|_2^2$. and transmits only the index $i$, requiring $\log_2 K$ bits per token. The codebook $\mathbf{e}$ can be learned via $K$-means clustering or updated online using exponential moving averages (Van Den Oord et al., 2017). This compact representation greatly reduces the communication cost between devices.

*Grouped VQ* (Yang et al., 2023) extends this idea by dividing the input vector into $G$ equal-length sub-vectors and quantizing each independently using separate codebooks. It increases expressiveness in the compressed representation and improves task accuracy, at the cost of higher communication overhead, $G \cdot \log_2 K$ bits per token. In experiments, ASTRA with Grouped VQ outperforms Vanilla VQ counterpart in accuracy, offering a tunable trade-off between bandwidth usage and model performance.

## 3 THE ASTRA FRAMEWORK

We present ASTRA, a communication-efficient multi-device inference framework for Transformer models. ASTRA is designed to minimize inter-device communication while preserving accuracy, enabling fast inference even in bandwidth-constrained environments. The framework achieves this through three key innovations: (1) Mixed-Precision Attention, (2) Noise-Augmented Vector Quantization, and (3) Distributed Class Tokens. We begin with an overview of the inference workflow, then describe each core design in detail.

### 3.1 OVERVIEW OF ASTRA

Figure 2 illustrates the inference procedure of the ASTRA framework. Given an input sequence consisting of a class token (optional) and $T$ content tokens, ASTRA first partitions the content tokens evenly across $N$ devices, assigning $T/N$ tokens to each device. To support classification and similar global tasks, the class token *CLS* is replicated to each device (*Distributed Class Token* in *Task Accuracy Preservation*). Each device thus holds a disjoint subset of the input sequence, along with its own class token copy, and maintains a full copy of the Transformer model.

Within each Transformer block, the inference proceeds in parallel across devices. Each device first applies our *Noise-Augmented Vector Quantization* (see *Task Accuracy Preservation*) to its local tokens, and transmits the corresponding low-bit indices to other devices. Each device now has access to the full input sequence, full-precision for local tokens, and vector quantized versions for non-local tokens. It then performs *Mixed-Precision Attention Computation* (see *Extreme Communication Compression*), computing attention maps over this hybrid set of representations. Since the feed-forward network (i.e., MLP) is position-wise independent, it is executed locally on each device without communication.

After all Transformer blocks, all class tokens, replicated across devices, are aggregated via average pooling to form a single unified representation. For classification tasks, this pooled class token is passed to a downstream prediction head to generate the final output. For generative tasks such as

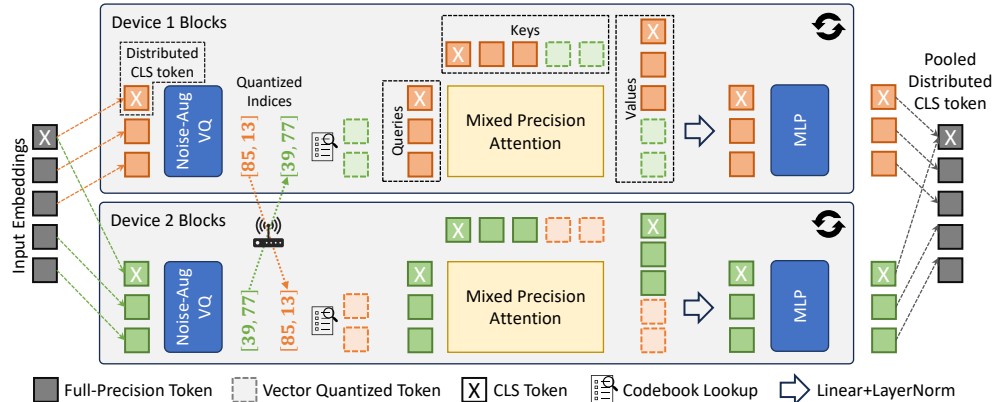

Figure 2: Overview of ASTRA with two devices. We introduce three key innovations: (1) Mixed-Precision Attention, (2) Noise-Augmented Vector Quantization, and (3) Distributed Class Tokens to achieve communication-efficient multi-device inference. ASTRA can be applied to transformers for both deterministic and generative tasks.

next-token prediction, there is no class token. Instead, the input sequence is evenly partitioned across devices for parallel encoding, which accelerates the initial digestion time during inference. Once the token encoding is complete, autoregressive decoding proceeds sequentially on a single device that holds the final (i.e., most recent) token in the input sequence.

## 3.2 EXTREME COMMUNICATION COMPRESSION

Transformer attention requires global context aggregation, which poses a communication bottleneck when the input tokens are partitioned across devices. In detail, the self-attention layers must use all tokens in the sequence, including those stored on other devices, to compute attention for each local token. As a result, each Transformer block must perform an all-to-all exchange of embeddings across devices, leading to significant communication overhead. For example, if the input tokens are evenly partitioned across $N$ devices, with each holding $T/N$ tokens, then transmitting full-precision embeddings requires sending $T/N \times D \times r$ bits per device per block, where $D$ is the hidden dimension and $r$ is the precision (e.g., float32). Such overhead becomes prohibitive under realistic bandwidth constraints.

**Mixed-Precision Attention.** To address this challenge, we propose to leverage both full-precision and compressed token representations during attention computation. Specifically, each device computes full-precision attention among its local tokens, and approximates attention interactions with non-local tokens (i.e., those stored on other devices) using vector-quantized embeddings. Only the low-bit indices of the vector-quantized embeddings are transmitted between devices.

Formally, for each local query $\mathbf{q}$, we compute attention over a mixed set of key and value pairs, full-precision representations from local tokens and vector-quantized representations from non-local tokens. Therefore, the attention map is computed as:

$$\text{Attn}(\mathbf{Q}, \mathbf{K}, \hat{\mathbf{K}}, \mathbf{V}, \hat{\mathbf{V}}) = \sigma\left(\frac{\mathbf{Q}[\mathbf{K} \mid \hat{\mathbf{K}}]^\top}{\sqrt{d_k}} \odot \mathbf{M}\right)[\mathbf{V} \mid \hat{\mathbf{V}}], \quad (1)$$

where $\hat{\mathbf{K}}$ and $\hat{\mathbf{V}}$ are derived from the vector-quantized embeddings $\hat{\mathbf{X}}$ via linear projections. The operator $\mid$ denotes row-wise concatenation, and $\sigma$ represents softmax operation. During the training stage, the attention mask $\mathbf{M}$ ensures that full-precision interactions are only applied between local tokens, while interactions with non-local tokens use their compressed counterparts.

**Vector-Quantized Non-Local Tokens.** The non-local compressed embeddings $\hat{\mathbf{X}}$ in Mixed-Precision Attention are produced by a vector quantization (VQ) module. Prior to transmission, each token embedding $\mathbf{X}$ is quantized by a codebook to an index $i$ via nearest-neighbor lookup. Since the codebook is shared across devices for each Transformer block, the receiving device can reconstruct $\hat{\mathbf{X}}$ using only the transmitted index $i$. This reduces the per-token communication cost from $rD$ bits to $\log_2 K$ bits, where $K$ is the codebook size, resulting in a compression ratio of $2457.6\times$ when $r = 32$, $D = 768$, and $K = 1024$.

The VQ module is jointly trained with the Transformer model. Specifically, the codebook is initialized by running $K$-means clustering over intermediate token embeddings from the pretrained model, and is further updated via exponential moving average during model fine-tuning. Following VQVAE (Van Den Oord et al., 2017), we apply a commitment loss to encourage token embeddings to stay close to their assigned centroids. The overall training objective is:

$$\mathcal{L} = \mathcal{L}_t + \beta \|\mathbf{X} - \text{sg}(\hat{\mathbf{X}})\|_2^2, \tag{2}$$

where $\mathcal{L}_t$ is the task loss, $\text{sg}(\cdot)$ denotes the stop-gradient operation, and $\beta$ controls the strength of the commitment term. This design encourages the model to align token embeddings with their corresponding quantized representations, which improves downstream task performance even under aggressive compression. Appendix F further empirically demonstrates that the commitment term and a well-configured loss weight is necessary for maintaining accuracy.

### 3.3 TASK ACCURACY PRESERVATION

**Noise-Augmented Vector Quantization.** Quantizing embeddings introduces discretization error, which can degrade model generalization. To mitigate this, we propose a novel regularization strategy called *Noise-Augmented Vector Quantization (NAVQ)*, which adds Gaussian noise to quantized embeddings during training. This technique is inspired by the Vicinal Risk Minimization (VRM) principle (Chapelle et al., 2000), which improves generalization by exposing the model to perturbations around each data point in input space. NAVQ extends this philosophy into the latent quantized embedding space. Instead of directly using the deterministic quantized embedding $\hat{\mathbf{X}}$ during training, we compute a noise-augmented version, $\tilde{\mathbf{X}} = \hat{\mathbf{X}} + \lambda \xi$, where $\lambda \in (0, 1]$ controls the noise magnitude and $\xi \sim \mathcal{N}(\boldsymbol{\mu}, \boldsymbol{\Sigma})$ is Gaussian noise sampled from the distribution of quantization residuals $\boldsymbol{\varepsilon} := \mathbf{X} - \hat{\mathbf{X}}$. This residual captures the error introduced by quantization, and the noise distribution is fit with empirical mean $\boldsymbol{\mu}$ and covariance $\boldsymbol{\Sigma}$ over training data.

By injecting noise into quantized embeddings during training, NAVQ restores a degree of continuity to the otherwise discrete latent space, encouraging the model to generalize across small perturbations and reducing sensitivity to codebook boundaries. At inference time, the noise is omitted and the model operates deterministically using $\hat{\mathbf{X}}$. We theoretically justify this approach in Appendix B and prove the following:

**Theorem 1** (Noise-Augmented Embeddings Improve Distributional Fidelity). *Let $\hat{\mathbf{X}}$ denote the quantized embedding of $\mathbf{X}$, and let $\tilde{\mathbf{X}} = \hat{\mathbf{X}} + \lambda \xi$ with $\xi \sim \mathcal{N}(\boldsymbol{\mu}, \boldsymbol{\Sigma})$ sampled from the quantization residuals. Then the 2-Wasserstein distance between the original embedding distribution $P_{\mathbf{X}}$ and the perturbed distribution $P_{\tilde{\mathbf{X}}}$ satisfies:*

$$W_2^2(P_{\mathbf{X}}, P_{\tilde{\mathbf{X}}}) < W_2^2(P_{\mathbf{X}}, P_{\hat{\mathbf{X}}}), \tag{3}$$

*i.e., the noise-augmented distribution is statistically closer to the true distribution than the raw quantized embedding.*

Empirically, NAVQ reduces overfitting and improves generalization. As shown in our ablation study (see Appendix F), setting $\lambda = 1.0$ improves validation accuracy by 0.86% compared to training without noise, demonstrating the effectiveness of this regularization under extreme compression.

**Distributed Class Tokens.** In Transformer-based classification models, such as ViT (Dosovitskiy et al., 2020), a special *class token* is prepended to the input sequence and used to aggregate information from all other tokens through attention. However, in the context of our Mixed-Precision Attention mechanism, attention between tokens on different devices is performed using vector-quantized embeddings. If the class token is assigned to a single device, it will attend to full-precision local tokens but only see vector-quantized representations from other devices. This asymmetric access to information introduces a bias in the class token's representation, potentially limiting its ability to effectively summarize the entire input sequence.

To address this issue, we introduce the *Distributed Class Token* mechanism. Instead of assigning the class token to a single device, we replicate it across all devices, creating one local copy per device. Each replica computes attention with full-precision local tokens and quantized non-local tokens. At the end of the model, all class token replicas are aggregated (e.g., via mean pooling) into a single vector, which is then passed to the final prediction head. This approach not only restores symmetry in

access to information but also reduces estimation error in the attention output, improving robustness to quantization artifacts. We formally justify this mechanism in Appendix C and prove the following:

**Theorem 2** (Variance Reduction via Distributed Class Tokens). *Let* $\mathbf{h}$ *denote the class token embedding of a full-precision global attention computation. Let* $\tilde{\mathbf{h}}_{\mathrm{single}}$ *be the output of a single-device class token using Mixed-Precision Attention, and let* $\tilde{\mathbf{h}}_{\mathrm{dist}}$ *be the average of* $N$ *distributed class token outputs. Then:*

$$\mathbb{E}\big[\|\tilde{\mathbf{h}}_{\mathrm{dist}} - \mathbf{h}\|_2^2\big] = \frac{1}{N}\mathbb{E}\big[\|\tilde{\mathbf{h}}_{\mathrm{single}} - \mathbf{h}\|_2^2\big], \tag{4}$$

*i.e., distributed class tokens reduce the expected attention output error by a factor of* $1/N$.

Empirically, our ablation study (see Appendix F) confirms that Distributed Class Tokens consistently outperform the single-token variant across all evaluated settings, yielding accuracy improvements between 0.37% and 7.13% depending on the compression level and commitment loss weight.

## 4 EMPIRICAL EVALUATION

This section evaluates the effectiveness of ASTRA by answering the following question: (1) Can ASTRA maintain model accuracy under aggressive token compression? (2) How much can ASTRA speed up inference under limited bandwidths compared to baselines? (3) How effective are the optimizations in ASTRA? We answer these questions through extensive experiments across Transformer models (ViT and GPT2), application domains (vision and NLP tasks), and deployment conditions (varying bandwidth, device count, compression settings, and device heterogeneity).

### 4.1 EXPERIMENTAL SETUP

**Environment.** ASTRA is implemented in PyTorch 2.5 and trained on a single L40S GPU with 40GB memory. For deployment, we simulate distributed inference on personal laptops provisioned with an NVIDIA 1660Ti GPU. We emulate a range of network conditions by enforcing bandwidth caps, enabling evaluation under constrained environments. Unless stated otherwise, experiments use 4 devices in a homogeneous setting. We report results under heterogeneous settings in Appendix D.

**Transformer Models.** We evaluate ASTRA across both encoder and decoder Transformer architectures: For encoder architecture, we focus on vision tasks with Vision Transformer (ViT-Base) (Dosovitskiy et al., 2020). For decoder architecture, we conduct experiments on NLP tasks with GPT2-Small (GPT2-S) and GPT2-Medium (GPT2-M) (Radford et al., 2019).

**Datasets and Metrics.** We evaluate ASTRA on both vision and language tasks. For vision, we use CIFAR-100 (Krizhevsky et al., 2009) and ImageNet-1K (Deng et al., 2009), reporting top-1 classification accuracy. For language modeling, we perform next-word prediction using two datasets, English Wikipedia and Wikitext-103 (Merity et al., 2016), and report perplexity (PPL; lower is better). The evaluation includes three settings: training and evaluating on Wikipedia (Foundation), training and evaluating on Wikitext-103, and a zero-shot evaluation where the model is trained on Wikipedia but directly evaluated on the Wikitext-103 validation set without further fine-tuning. The last zero-shot setting follows the evaluation used in the original GPT2 (Radford et al., 2019) and serves to assess the model's generalization to unseen domains. All experiments are conducted with a fixed random seed (42) for reproducibility. To demonstrate the robustness of ASTRA across different runs, results averaged over multiple seeds are reported in Appendix D. The memory cost analysis is included in Appendix G.

**Baselines.** We compare ASTRA with both single-device and three multi-device inference approaches.

- **Original Model**: The baseline model runs entirely on a single device using float32 precision. We compare with the float32 model as ASTRA builds on top of this model for a fair comparison. Later, we show ASTRA can be combined with model quantization.
- **Tensor Parallelism (TP)**: Represented by Megatron-LM (Shoeybi et al., 2019), which partitions weight matrices across devices and requires two allreduce operations per Transformer layer.
- **Sequence Parallelism (SP)**: Introduced by Voltage (Hu & Li, 2024), which partitions the input sequence and performs one AllGather operation per layer.
- **Block Parallelism (BP)**: Proposed by DeTransformer (Du et al., 2024), which replaces Transformer blocks with multiple parallel sublayers for distributed execution.

For BP, we evaluate two most efficient design variants proposed in (Du et al., 2024): (i) **BP+AllGather (BP+AG)** minimizes communication by performing more local computation, and (ii) **BP+SequenceParallel (BP+SP)** reduces local computation at the cost of moderate communication overhead. Both variants include a hyperparameter $N_b$ that controls the number of original Transformer blocks retained. A smaller $N_b$ leads to fewer communications and thus lower latency. We compare against BP configurations with $N_b = 1$ and $N_b = 4$.

**ASTRA Settings.** For the Noise-Augmented Vector Quantization in ASTRA, the codebook size is 1024, representing each transmitted token with 10 bits (i.e., $\log_2 1024$). The noise magnitude $\lambda$ is 1.0 in the main results and we test other settings including $\lambda \in \{0.0, 0.1, 0.3\}$ in Appendix F. We further evaluate the use of *Grouped VQ*, as introduced in *Background*, which splits each input vector into $G$ groups and applies vector quantization independently to each group using separate codebooks. We experiment with group sizes of 16 and 32, in addition to *Vanilla VQ* of a single group. Increasing the number of groups leads to a higher bits per token and thus reduces the overall compression ratio proportionally.

We load the pre-trained weights for all the transformer models from the HuggingFace official model zoo. Then ASTRA is fine-tuned for additional iterations on each dataset using the Adam optimizer (Kingma & Ba, 2014). Specifically, for vision tasks, ASTRA is fine-tuned on CIFAR-100 and ImageNet-1K for 32 and 4 epochs, respectively. For NLP tasks, ASTRA is fine-tuned on 1 million samples from English Wikipedia and the complete Wikitext-103 dataset for 1 epoch. During fine-tuning, we test with different commitment loss weights $\beta \in \{0.0001, 0.0002, 0.0005\}$ in Appendix F and report the best accuracy performance in *Results on Accuracy*.

## 4.2 RESULTS ON ACCURACY AND COMMUNICATION COSTS

We evaluate the accuracy of ASTRA across three Transformers, ViT-Base, GPT2-S, and GPT2-M, on vision and NLP benchmarks. Note that we only report the baseline accuracy for the original model. Existing multi-device baselines, including Megatron-LM (Shoeybi et al., 2019) and Voltage (Hu & Li, 2024), do not incur any accuracy loss since they merely reorganize computation without altering the model's numerical outputs. Therefore, their results are equivalent to the original model and are omitted here for clarity. Alongside accuracy, we also report the associated communication overhead, measured as the total amount of data exchanged per token during a single forward pass (i.e., *Total Bits per Token*). Results are summarized in Tables 1 and 3.

**ViT-Base.** ASTRA maintains high accuracy on ViT-Base for image classification (CIFAR-100 and ImageNet-1K), with less than 3.58% degradation even under 2457.6× compression, as shown in Table 1. With 32 groups, ASTRA achieves 91.64% on CIFAR-100 and 80.28% on ImageNet-1K, closely matching the original performance of 92.53% and 80.32%. To further assess scalability, we fix the compres-

Table 1: Task accuracy and communication overhead on **CIFAR-100** and **ImageNet-1K** with **ViT-Base**.

| Model | #Groups | Total Bits per Token | Compression Ratio | CIFAR-100 | ImageNet |
|---|---|---|---|---|---|
| ViT-Base | - | 294912 | - | 92.53 | 80.32 |
| **ASTRA** | 1 | 120 | 2457.6 | 88.95 | 77.39 |
| | 16 | 1920 | 153.6 | 90.77 | 78.80 |
| | 32 | 3840 | 76.8 | **91.64** | **80.28** |

sion configuration (32 groups) and evaluate ASTRA on CIFAR-100 using varying numbers of devices. As shown in Table 2, ASTRA preserves model accuracy within 1.39% of the original model across different device counts.

**GPT2.** Table 3 summarizes the perplexity (PPL) results of ASTRA on the next-token prediction task, i.e., Wikipedia and Wikitext-103, using GPT2-S and GPT2-M. Overall, ASTRA achieves competitive performance under aggressive communication compression. Note that perplexity is an exponential function of the language modeling loss, i.e., $\text{PPL} = \exp(\mathcal{L})$, and thus small dif-

Table 2: Accuracy of ASTRA on CIFAR-100 under different numbers of devices.

| Model | ViT-Base | ASTRA (#Groups = 32) | | | |
|---|---|---|---|---|---|
| #Devices | 1 | 2 | 4 | 6 | 8 |
| Accuracy | 92.53 | 91.86 | 91.64 | 91.35 | 91.14 |

ferences in loss can result in amplified changes in PPL. Specifically, on GPT2-M, the PPL on Wikitext-103 increases from 14.8 (loss = 2.70) to 16.84 (loss = 2.82), reflecting only a 4.4% increase in loss despite a 102.4× compression ratio. Similarly, on Wikipedia, the loss increases marginally from 2.5 to 2.63 (PPL from 12.16 to 13.83), confirming that much of the accuracy is preserved under significant transmission savings.

**Zero-Shot Generalization.** We also evaluate ASTRA in the zero-shot setting by directly evaluating the model trained with Wikipedia on the Wikitext-103 validation set. Here, we observe a larger performance drop compared to the original model. For example, GPT2-M's zero-shot PPL rises from 43.22 to 62.29 with AS-TRA at 32 groups. This performance drop suggests a limitation of ASTRA in zero-shot generalization: the discretization introduced by VQ reduces the diversity of token representations and hinders out-of-distribution data generalization.

Table 3: Task performance (i.e., perplexity) and communication overhead on **Wikipedia** and **Wikitext-103** with **GPT2**.

| Model | #Groups | Bits per Token | Compression Ratio | Wikipedia | Wikitext-103 Fine-Tuned | Wikitext-103 Zero-Shot |
|---|---|---|---|---|---|---|
| GPT2 - S | - | 294912 | - | 15.79 | 18.96 | 58.91 |
| ASTRA | 1 | 120 | 2457.6 | 21.46 | 25.98 | 120.7 |
| | 16 | 1920 | 153.6 | 18.84 | 23.22 | 94.8 |
| | 32 | 3840 | 76.8 | **17.39** | **20.95** | **76.24** |
| GPT2 - M | - | 786432 | - | 12.16 | 14.8 | 43.22 |
| ASTRA | 1 | 240 | 3276.8 | 17.86 | 21.97 | 96.99 |
| | 16 | 3840 | 204.8 | 14.43 | 18.03 | 75.29 |
| | 32 | 7680 | 102.4 | **13.83** | **16.84** | **62.29** |

**Heterogeneous Devices.** In heterogeneous settings where devices have different compute capacities, ASTRA can adapt by assigning more tokens to stronger devices. Our training uses a randomized token-to-device mapping to learn a unified codebook, enabling direct generalization to unseen heterogeneity without retraining. Experiments on ImageNet-1K with 4 devices show that ASTRA maintains within 1.43% accuracy drop compared to the original ViT-Base. We further observe that higher heterogeneity increases the full-precision attention rate, leading to better accuracy (See Appendix D for details).

### 4.3 RESULTS ON INFERENCE LATENCY

We report latency improvement using a 12-layer Transformer encoder with 768 hidden dimensions. We compare ASTRA with existing multi-device inference baselines and evaluate their latency across three key dimensions, varying bandwidth, device count, and input token length, in Figure 1, 4, and 5.

**Varying Bandwidth.** Figure 1 presents the speedup of multi-device methods over single-device inference, evaluated across inter-device bandwidths ranging from 10 Mbps to 500 Mbps. We fix the number of devices to 4 and the input token length to 1024. Additional device count and sequence length configurations are provided in Appendix E. Across all bandwidths, ASTRA consistently outperforms all baselines and maintains substantial speedup, even under extremely limited bandwidth. For instance, ASTRA achieves a speedup of $1.27 - 2.74\times$ at 20 Mbps, while all other baselines perform even worse than single-device inference. Even at 10 Mbps, our method with 16 and 1 quantization groups still delivers $1.26 - 2.65\times$ speedup, demonstrating strong scalability to bandwidth bottlenecks.

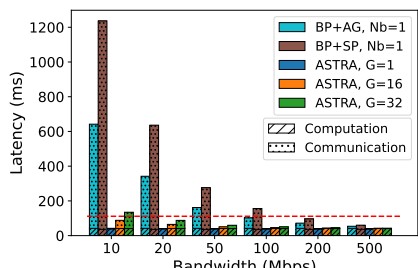

Figure 3: Latency breakdown of local computation and inter-device communication time. The red dashed line represents the single-device latency.

We also visualize the absolute latency breakdown in Figure 3. Specifically, we depict the latency breakdown for the two fastest baselines, BP+AG and BP+SP when $N_b = 1$, as well as ASTRA with different groups. We can see that the communication time dominates in total latency for BP+AG and BP+SP, accounting for as much as $58.55 - 93.47\%$ of total runtime at low-bandwidth settings below 100 Mbps. In contrast, ASTRA effectively mitigates this communication bottleneck via aggressive compression, thereby significantly reducing total latency.

We further summarize the relative speedup of ASTRA over each baseline across different bandwidth in Table 4. ASTRA's advantage becomes more significant under stricter bandwidth. The speedup of ASTRA over Sequence Parallelism (SP) reflects the benefit of our Mixed-Precision Attention, contributing up to $171.82\times$ latency reduction under low-bandwidth settings.

Table 4: ASTRA's speedup over baseline methods on 4 devices with 1024 tokens.

| Bandwidth (Mbps) | 10 | 20 | 50 | 100 | 200 | 500 |
|---|---|---|---|---|---|---|
| TP | 342.74 | 177.89 | 73.14 | 37.19 | 19.02 | 8.05 |
| SP | 171.82 | 89.41 | 37.05 | 19.08 | 9.99 | 4.51 |
| BP+AG, Nb=1 | 15.25 | 8.41 | 4.07 | 2.58 | 1.83 | 1.37 |
| BP+SP, Nb=1 | 29.37 | 15.66 | 6.95 | 3.96 | 2.45 | 1.53 |

**Varying Device Counts.** Figure 4 shows the latency speedup comparison as the number of devices increases from 2 to 8. We fix the input token length to 1024 and illustrate two representative

bandwidth settings, 20 Mbps and 200 Mbps (more results see Appendix E). For both bandwidth, ASTRA consistently achieves higher speedup than all baselines. As the number of devices increases, more computation can be parallelized, leading to greater latency reduction. For example, under 20 Mbps, ASTRA with 1 group improves from $1.72\times$ speedup with 2 devices to $3.69\times$ with 8 devices.

**Varying Input Token Length.** Figure 5 presents the latency speedup comparison as the input token length increases from 256 to 4096. Similarly, we fix the number of devices to 4 and evaluate under 20 Mbps and 200 Mbps (more bandwidth see Appendix E). Across all sequence lengths, ASTRA consistently outperforms existing methods and our superiority becomes more significant at longer input lengths. In real-world applications, longer input token lengths tend to form a more substantial barrier to achieving low-latency inference. At 512 tokens and 20 Mbps bandwidth, for instance, ASTRA achieves a latency speedup of $1.98\times$ compared to the fastest baseline BP-AG of $0.25\times$, highlighting the practical value of ASTRA in real deployment scenarios.

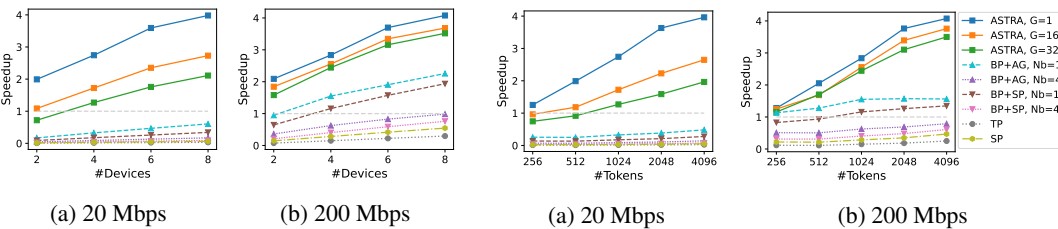

| (a) 20 Mbps | (b) 200 Mbps | (a) 20 Mbps | (b) 200 Mbps |

Figure 4: Speedup comparison under different numbers of devices (w/ 1024 tokens).

Figure 5: Speedup comparison under different input token length (w/ 4 devices).

### 4.4 COMPATIBILITY WITH BIT QUANTIZATION

To demonstrate the compatibility of ASTRA with model compression, we apply post-training quantization to the standard ViT-Base model and our ASTRA variants, and evaluate their performance on ImageNet-1K under 8-bit and 4-bit settings. Table 5 summarizes the accuracy and latency results, with latency measured under 200 Mbps bandwidth, using 4 devices and an input token length of 1024.

**Accuracy.** 8-bit and 4-bit quantization yield minimal accuracy degradation. When ASTRA is combined with bit quantization, performance remains robust. For instance, applying 4-bit quantization to ASTRA with 32 groups still achieves 79.78% accuracy. This supports the claim that ASTRA can be layered on top of bit quantization methods to further reduce latency while preserving task performance.

Table 5: Accuracy and latency of ViT-Base and ASTRA on ImageNet-1K under different precision (FP32, 8-bit, 4-bit).

| Model | | Accuracy | | | Latency (ms) $_{\text{Speedup}}$ | | |
| Name | #Group | FP32 | 8-bit | 4-bit | FP32 | 8-bit | 4-bit |
|---|---|---|---|---|---|---|---|
| ViT-Base | - | 80.32 | 80.27 | 80.19 | 99.9 | 79.8 | 103.2 |
| ASTRA | 1 | 77.39 | 77.32 | 76.82 | 36.7 $_{2.73\times}$ | 50.6 $_{1.58\times}$ | 44.6 $_{2.31\times}$ |
| | 16 | 78.80 | 78.76 | 78.43 | 41.0 $_{2.44\times}$ | 51.7 $_{1.54\times}$ | 50.2 $_{2.06\times}$ |
| | 32 | **80.28** | **80.26** | **79.78** | 44.5 $_{2.25\times}$ | 59.3 $_{1.35\times}$ | 56.9 $_{1.81\times}$ |

**Latency.** Combining ASTRA with quantization pushes end-to-end Transformer acceleration beyond either method alone. For instance, the 4-bit ASTRA on 4 devices can achieve $1.81 - 2.31\times$ speedup over 4-bit ViT-Base on a single device. Notice that the actual speedup from bit quantization depends on kernel implementation, hardware-specific optimization, and target device. In some cases, e.g., 4-bit ViT-Base, it may even slow down due to conversion or kernel overhead.

### 4.5 SCALABILITY TO LARGE TRANSFORMER MODELS

To evaluate the scalability of ASTRA to large language models, we experiment with Llama-3-8B (Dubey et al., 2024) for next-token prediction on the English Wikipedia dataset. 8-bit quantization is enabled for all methods, including the baselines and ASTRA, to execute inference with NVIDIA TitanX GPUs and keep fair comparisons. When evaluating the latency, we fixed the input token length to 1024 using 4 devices.

**Accuracy.** Table 6 reports perplexity (PPL, lower is better) together with the communication cost in bits per token as we vary the number of groups in ASTRA. Compared to the single-device Llama-3-8B baseline, ASTRA maintains performance close to the original while significantly reducing communication. For example, when the number of groups is 1, ASTRA incurs only a small increase in PPL

from 5.81 (loss = 1.76) to 7.73 (loss = 2.04), while achieving a $1600\times$ reduction in communication, confirming that the proposed multi-device inference framework can scale to 8B-parameter models.

Table 6: Task performance (i.e., perplexity) and communication overhead on **Wikipedia** with **Llama-3-8B**, with and without 5% packet loss.

| Model | #Groups | Bits per Token | Compression Ratio | PPL | PPL under 5% packet loss |
|---|---|---|---|---|---|
| Llama-3-8B | - | 1,048,576 | - | 5.8118 | - |
| **ASTRA** | 1 | 640 | 1,638.4 | 7.7336 | 7.7294 |
| | 16 | 10,240 | 102.4 | 7.5879 | 7.5900 |
| | 32 | 20,480 | 51.2 | 7.4360 | 7.4431 |

Table 7: Latency (s) comparison between ASTRA and baselines across different bandwidths (Mbps).

| Bandwidth (Mbps) | 10 | 20 | 50 | 100 | 200 | 500 |
|---|---|---|---|---|---|---|
| Llama-3-8B | | | 4.578 | | | |
| TP | 430.952 | 216.291 | 87.449 | 44.499 | 23.025 | 10.140 |
| SP | 28.256 | 14.939 | 6.888 | 4.215 | 2.857 | 2.052 |
| BP, Nb=4 | 4.642 | 3.047 | 2.085 | 1.753 | 1.586 | **1.485** |
| BP, Nb=8 | 8.011 | 4.780 | 2.773 | 2.101 | 1.762 | 1.561 |
| ASTRA, G=1 | **1.563** | **1.549** | **1.547** | **1.545** | **1.541** | 1.540 |
| ASTRA, G=16 | 1.661 | 1.659 | 1.595 | 1.572 | 1.559 | 1.548 |
| ASTRA, G=32 | 1.940 | 1.796 | 1.661 | 1.630 | 1.603 | 1.583 |

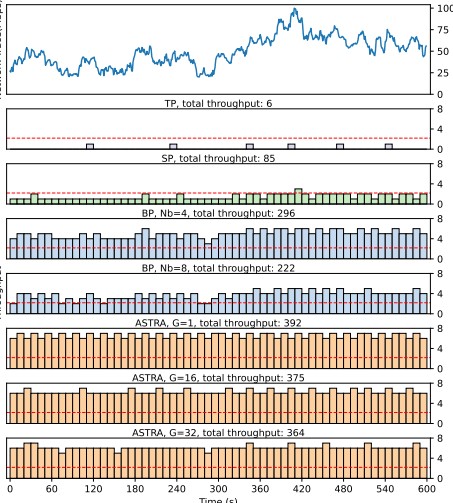

Figure 6: Overall request throughput comparison under dynamic network bandwidth with a fixed 600-second trace. Red dashed line represents the single-device baseline.

**Latency.** Table 7 summarizes end-to-end latency under varying bandwidth ranging from 10 to 500 Mbps. ASTRA consistently achieves lower latency than state-of-the-art multi-device baselines at low bandwidth (e.g., 10–100 Mbps). Specifically, ASTRA attains $1.13 - 5.13\times$ speedup over the fastest baseline, i.e., BP, Nb=4. Because 8-bit quantization is uniformly applied to all methods, these gains isolate the benefit of ASTRA 's communication-efficient design.

**Non-Ideal Network Conditions.** We further stress the inter-device network with packet loss and time-varying bandwidth. For the packet loss, since WiFi networks generally experience packet loss rates of around 1% to 5% (Sheshadri & Koutsonikolas, 2017), Table 6 includes perplexity when we inject a 5% random packet loss rate without retransmission. ASTRA preserves task performance under this moderate packet loss, showing only minor degradation in terms of PPL.

We further simulate fluctuating network conditions using synthetic bandwidth traces generated by a Markovian model from Pensieve (Mao et al., 2017), where each state corresponds to a bandwidth between 20–100 Mbps, and transitions are biased toward nearby states to capture temporal correlation. Figure 6 depicts the 600-second bandwidth trace together with the overall resolved requests when using a single batch size for the single-device baseline (i.e., the red dashed line) and multi-device methods on 4 devices (i.e., the bar charts). For each method, the bars illustrate the number of resolved requests every 10 seconds, and the overall throughput is reported in the title. Under this fluctuating bandwidth, ASTRA delivers higher throughput than both single-device inference and multi-device baselines, demonstrating that the proposed communication-efficient mechanisms remain effective under realistic, non-ideal network conditions.

## 5 CONCLUSION

We present ASTRA, a communication-efficient framework for accelerating Transformer inference in multi-device settings. By integrating sequence parallelism with a novel Mixed-Precision Attention mechanism, ASTRA significantly reduces inter-device communication while preserving accuracy. Extensive experiments across vision and NLP tasks demonstrate that ASTRA delivers substantial end-to-end latency improvements over existing baselines, achieving up to $2.64\times$ speedup over single-device inference and up to $15.25\times$ over state-of-the-art multi-device methods, under constrained bandwidth as low as 10 Mbps. Our results highlight the potential of ASTRA for practical deployment of Transformer models in real-world, bandwidth-limited environments.

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

## A  ADDITIONAL RELATED WORK

**Deploying Transformers on Edge Devices.** Substantial efforts have been devoted to enabling Transformer models on edge devices through model compression and architecture simplification. For example, Michel et al. (Michel et al., 2019) proposed pruning attention heads to reduce computational cost, while Q8BERT (Zafrir et al., 2019) quantizes BERT weights from 32-bit to 8-bit to accommodate memory-limited environments. Other approaches, such as parameter factorization in ALBERT (Lan et al., 2020) and knowledge distillation methods (Lin et al., 2022), aim to construct lightweight variants of Transformer architectures suitable for resource-constrained hardware. These techniques focus on discovering a compact model that maintains acceptable task performance under tight latency or memory budgets.

In contrast, ASTRA targets distributed inference while preserving the original model architecture. The compressed transformer models from the above techniques can also leverage ASTRA 's distributed inference system for further acceleration, as long as they retain the core transformer architecture. This makes ASTRA an orthogonal solution that offers further performance improvements without requiring re-design or re-training for new hardware targets.

**Distributed Inference Systems.** Distributed inference has emerged as a practical strategy to accelerate computation. Early works such as DeepThings (Zhao et al., 2018) exploited the partial receptive fields of convolutional neural networks (CNNs) to parallelize inference by splitting intermediate feature maps across multiple devices. The follow-up works, including CoEdge (Zeng et al., 2020), DeepSlicing (Zhang et al., 2021), and EdgeFlow (Hu & Li, 2022), further incorporated network heterogeneity and device resource profiling to optimize system throughput. However, these methods are designed specifically for CNN-based models and are not applicable to the self-attention structure in Transformers.

Recent work has begun to explore multi-device inference for Transformers by adapting techniques from distributed training. PipeEdge (Hu et al., 2022) utilizes pipeline parallelism to improve throughput, but its efficiency relies on large batch sizes and does not benefit per-request latency. Other systems, such as DeepSpeed (Aminabadi et al., 2022) and Megatron-LM (Shoeybi et al., 2019) apply tensor parallelism by splitting weight matrices across devices, which leads to frequent and expensive inter-device communication. To reduce this cost, Voltage (Hu & Li, 2024) introduces sequence parallelism by distributing input tokens across devices and minimizing the number of cross-device interactions per Transformer block. Galaxy (Ye et al., 2024) further combines tensor and sequence parallelism, while DeTransformer (Du et al., 2024) even modifies the Transformer block structure to enable more efficient distribution.

Despite their contributions, these methods still rely on high-bandwidth connections to achieve meaningful speedups. In contrast, ASTRA significantly reduces the required bandwidth to only 10 Mbps, while still achieving $2.64\times$ end-to-end latency speedup, making it far more practical for real-world deployments in constrained edge environments.

## B  PROOF FOR NOISE-AUGMENTED VECTOR QUANTIZATION

**Theorem 1** (Noise-Augmented Embeddings Improve Distributional Fidelity)**.** *Let* $\hat{\mathbf{X}}$ *denote the quantized embedding of* $\mathbf{X}$*, and let* $\tilde{\mathbf{X}} = \hat{\mathbf{X}} + \lambda \xi$ *with* $\xi \sim \mathcal{N}(\boldsymbol{\mu}, \boldsymbol{\Sigma})$ *sampled from the quantization residuals. Then the 2-Wasserstein distance between the original embedding distribution* $P_{\mathbf{X}}$ *and the perturbed distribution* $P_{\tilde{\mathbf{X}}}$ *satisfies:*

$$W_2^2(P_{\mathbf{X}}, P_{\tilde{\mathbf{X}}}) < W_2^2(P_{\mathbf{X}}, P_{\hat{\mathbf{X}}}), \tag{5}$$

*i.e., the noise-augmented distribution is statistically closer to the true distribution than the raw quantized embedding.*

*Proof of Theorem B.* Let

$$\mathbf{m}_X = \mathbb{E}[\mathbf{X}], \quad \mathbf{m}_{\hat{X}} = \mathbb{E}[\hat{\mathbf{X}}], \quad \mathbf{m}_{\tilde{X}} = \mathbb{E}[\tilde{\mathbf{X}}], \tag{6}$$

and let $C_X$, $C_{\hat{X}}$, $C_{\tilde{X}}$ denote the corresponding covariance matrices. By definition of the quantization error $\boldsymbol{\varepsilon} = \mathbf{X} - \hat{\mathbf{X}}$ we have $\mathbf{m}_X = \mathbf{m}_{\hat{X}} + \mu$ and $C_X = C_{\hat{X}} + \Sigma$. As the injected noise $\xi \sim \mathcal{N}(\mu, \Sigma)$,,

we know that

$$\mathbf{m}_{\tilde{X}} = \mathbf{m}_{\hat{X}} + \lambda\mu, \quad C_{\tilde{X}} = C_{\hat{X}} + \lambda^2\Sigma. \tag{7}$$

The Wasserstein distance between two Gaussian distributions can be computed by

$$W_2^2(P_1, P_2) = \|m_1 - m_2\|_2^2$$
$$+ \operatorname{trace}\left(C_1 + C_2 - 2\left(C_2^{\frac{1}{2}} C_1 C_2^{\frac{1}{2}}\right)^{\frac{1}{2}}\right) \tag{8}$$
$$= \|m_1 - m_2\|_2^2 + d_B^2(C1, C2),$$

where $m$ and $C$ are mean and covariance of the distributions, $d_B$ is the Bures metric.

To prove $W_2^2(P_{\mathbf{X}}, P_{\tilde{\mathbf{X}}}) < W_2^2(P_{\mathbf{X}}, P_{\hat{\mathbf{X}}})$, we will first show the mean term of $\tilde{\mathbf{X}}$ is smaller, then the Bures term is smaller.

**Step 1 (mean term is smaller).** Using the mean part of the Gaussian $W_2$ formula, we have

$$\|\mathbf{m}_X - \mathbf{m}_{\hat{X}}\|_2^2 - \|\mathbf{m}_X - \mathbf{m}_{\tilde{X}}\|_2^2$$
$$= \|\mu\|_2^2 - (1-\lambda)^2\|\mu\|_2^2 \tag{9}$$
$$= (2\lambda - \lambda^2)\|\mu\|_2^2 > 0,$$

because $0 < \lambda \leq 1$. Then we prove that

$$\|\mathbf{m}_X - \mathbf{m}_{\tilde{X}}\|_2^2 < \|\mathbf{m}_X - \mathbf{m}_{\hat{X}}\|_2^2 \tag{10}$$

**Step 2 (Bures term is smaller).**

For analytical clarity, we assume that the quantization errors $\varepsilon$ are independent and identically distributed across dimensions, i.e. $\varepsilon_k \overset{\text{i.i.d.}}{\sim} \mathcal{N}(0, \sigma^2)$. Under this assumption, the global error covariance becomes $\Sigma = \sigma^2 I$, which commutes with $C_{\hat{X}}$, namely $C_{\hat{X}}\Sigma = \Sigma C_{\hat{X}}$. Consequently, $C_{\hat{X}}$ and $\Sigma$ can be diagonalized by the same orthonormal eigenbasis $U$. Then we have

$$U^\top C_{\hat{X}} U = \operatorname{diag}(\sigma_{\hat{X},i}^2),$$
$$U^\top \Sigma U = \operatorname{diag}(\sigma_i^2), \tag{11}$$
$$\sigma_{\hat{X},i}, \sigma_i \geq 0,$$

then we obtain

$$U^\top C_X U = \operatorname{diag}(\sigma_{X,i}^2), \ \sigma_{X,i}^2 = \sigma_{\hat{X},i}^2 + \sigma_i^2,$$
$$U^\top C_{\tilde{X}} U = \operatorname{diag}(\sigma_{\tilde{X},i}^2), \ \sigma_{\tilde{X},i}^2 = \sigma_{\hat{X},i}^2 + \lambda^2\sigma_i^2. \tag{12}$$

For diagonal matrices the Bures term in the $W_2$ expression reduces to a sum of squared differences of *standard deviations*:

$$d_B^2(C_A, C_B) = \sum_i \left(\sigma_{A,i} - \sigma_{B,i}\right)^2. \tag{13}$$

Then we have

$$d_B^2(C_X, C_{\hat{X}}) - d_B^2(C_X, C_{\tilde{X}})$$
$$= \sum_i \left((\sigma_{X,i} - \sigma_{\hat{X},i})^2 - (\sigma_{X,i} - \sigma_{\tilde{X},i})^2\right) \tag{14}$$
$$= \sum_i (\sigma_{\tilde{X},i} - \sigma_{\hat{X},i})\left(2\sigma_{X,i} - (\sigma_{\tilde{X},i} + \sigma_{\hat{X},i})\right).$$

Because $\sigma_{\hat{X},i} \leq \sigma_{\tilde{X},i} \leq \sigma_{X,i}$, we have $\sigma_{\tilde{X},i} - \sigma_{\hat{X},i} > 0$ and $2\sigma_{X,i} - (\sigma_{\tilde{X},i} + \sigma_{\hat{X},i}) > 0$, therefore Equation 14 is positive. Thus, we prove that

$$d_B^2(C_X, C_{\tilde{X}}) < d_B^2(C_X, C_{\hat{X}}). \tag{15}$$

**Summary.** Since both the mean part and the Bures part are strictly smaller for $(\mathbf{m}_{\tilde{X}}, C_{\tilde{X}})$ than for $(\mathbf{m}_{\hat{X}}, C_{\hat{X}})$, hence we have completed the proof for

$$W_2^2(P_{\mathbf{X}}, P_{\tilde{\mathbf{X}}}) < W_2^2(P_{\mathbf{X}}, P_{\hat{\mathbf{X}}}), \tag{16}$$

which completes the proof. $\square$

## C    PROOF FOR DISTRIBUTED CLASS TOKENS

**Theorem 2** (Variance Reduction via Distributed Class Tokens). *Let $\mathbf{h}$ denote the class token embedding of a full-precision global attention computation. Let $\tilde{\mathbf{h}}_{\text{single}}$ be the output of a single-device class token using Mixed-Precision Attention, and let $\tilde{\mathbf{h}}_{\text{dist}}$ be the average of $N$ distributed class token outputs. Then:*

$$\mathbb{E}\big[\|\tilde{\mathbf{h}}_{\text{dist}} - \mathbf{h}\|_2^2\big] = \frac{1}{N}\mathbb{E}\big[\|\tilde{\mathbf{h}}_{\text{single}} - \mathbf{h}\|_2^2\big], \tag{17}$$

*i.e., distributed class tokens reduce the expected attention output error by a factor of $1/N$.*

*Proof of Theorem 2.*

**Setup.** Tokens are evenly partitioned: $\mathbf{X} = \bigcup_{i=1}^{N} \mathbf{X}^{(i)}$, $|\mathbf{X}^{(i)}| = T/N$. Each device stores local keys $\mathbf{k}_j$ and values $\mathbf{v}_j$ in full precision for $j \in \mathbf{X}^{(i)}$, and transmits quantized versions $\tilde{\mathbf{k}}_j = \mathbf{k}_j + \delta\mathbf{k}_j$ and $\tilde{\mathbf{v}}_j = \mathbf{v}_j + \delta\mathbf{v}_j$ to other devices, where $\delta\mathbf{k}_j$ and $\delta\mathbf{v}_j$ are the error introduced by quantization. For every non-local token, we assume

$$\begin{aligned}
\mathbb{E}[\delta\mathbf{k}_j] &= \mathbf{0}, \\
\mathbb{E}[\delta\mathbf{v}_j] &= \mathbf{0}, \\
\text{Cov}(\delta\mathbf{k}_j) &= \sigma_k^2\mathbf{I}, \\
\text{Cov}(\delta\mathbf{v}_j) &= \sigma_v^2\mathbf{I},
\end{aligned} \tag{18}$$

and errors are mutually independent.

The variances $\sigma_k^2, \sigma_v^2$ are bounded according to the classical high-rate VQ theory (Zador, 1982; Gersho & Gray, 1991). It shows that, under mild assumptions on the feature distribution, the mean-squared quantization error of an optimal $K$-level VQ in dimension $d$ satisfies

$$\mathbb{E}\|\mathbf{X} - \hat{\mathbf{X}}\|_2^2 \le C_d \cdot \sigma_{\mathbf{X}}^2 \cdot K^{-2/d}, \tag{19}$$

where $\hat{\mathbf{X}}$ denotes the quantized embedding of $\mathbf{X}$, and $C_d$ is a constant depending on the dimension $d$. This implies a per-dimension variance bound

$$\sigma_k^2, \sigma_v^2 = \frac{1}{d}\mathbb{E}\|\mathbf{X} - \hat{\mathbf{X}}\|_2^2 \le \tilde{C} \cdot \sigma_{\mathbf{X}}^2 \cdot K^{-2/d}. \tag{20}$$

**Full-Precision Attention.** For a query $\mathbf{q}$, the attention logits are $a_j = \mathbf{q}^\top\mathbf{k}_j/\sqrt{d}$, attention weights $\alpha_j = \text{softmax}(a_j)$, and the output $\mathbf{h} = \sum_{j=1}^{T} \alpha_j\,\mathbf{v}_j$.

**Mixed-Precision Attention via First-Order Taylor Expansion.** For a non-local token, the logit is perturbed by the key noise,

$$\tilde{a}_j = \frac{\mathbf{q}^\top(\mathbf{k}_j + \delta\mathbf{k}_j)}{\sqrt{d}} = a_j + \underbrace{\frac{\mathbf{q}^\top\delta\mathbf{k}_j}{\sqrt{d}}}_{=:e_j^{(k)}}. \tag{21}$$

Because $e_j^{(k)}$ is small, we could linearise the softmax function by first-order Taylor expansion. Specifically, the softmax function is $\alpha_j = \exp(a_j)/\sum_j \exp(a_j)$, and its Jacobian is $\partial\alpha_j/\partial a_k = \alpha_j(\delta_{jk} - \alpha_k)$, where $\delta_{jk}$ is the Kronecker delta. Therefore, we have the perturbed attention weights,

$$\begin{aligned}
\tilde{\alpha}_j &\approx \alpha_j + \sum_k \alpha_j(\delta_{jk} - \alpha_k)\,e_k^{(k)} \\
&= \alpha_j \; + \; \alpha_j\Big(e_j^{(k)} - \sum_k \alpha_k e_k^{(k)}\Big) \\
&=: \alpha_j + \delta\alpha_j.
\end{aligned} \tag{22}$$

The terms $\delta\alpha_j$ remain zero-mean and mutually independent.

Then the mixed-precision output is

$$\tilde{\mathbf{h}} = \sum_j \tilde{\alpha}_j\,(\mathbf{v}_j + \delta\mathbf{v}_j), \tag{23}$$

where $\delta\mathbf{v}_j = \mathbf{0}$ if $j$ is the local token index.

**Attention Output Error.** Subtracting $\mathbf{h}$ and discarding higher-order noise products, the first-order output error is

$$
\begin{aligned}
\boldsymbol{\delta} &:= \tilde{\mathbf{h}} - \mathbf{h} \\
&= \sum_{\text{non-local } j} \big(\alpha_j\, \delta\mathbf{v}_j + \delta\alpha_j\, \mathbf{v}_j\big) \\
&= \underbrace{\sum_j \alpha_j\, \delta\mathbf{v}_j}_{\text{V-error: } \mathbf{e}^{(v)}} + \underbrace{\sum_j \alpha_j\big(e_j^{(k)} - \sum_k \alpha_k\, e_k^{(k)}\big)\mathbf{v}_j}_{\text{K-propagated error: } \mathbf{e}^{(k)}}.
\end{aligned}
\tag{24}
$$

*Both error components are zero-mean random vectors and each coordinate has variance bounded by $C_1\sigma_v^2 + C_2\sigma_k^2$, where $C_1$ and $C_2$ are deterministic constants.*

Specifically, for the first value-error component $\mathbf{e}^{(v)} = \sum_{j\in\mathcal{N}} \alpha_j\, \delta\mathbf{v}_j$, where $\mathcal{N}$ is the set of $m = \frac{N-1}{N}T$ non-local tokens per device and each $\delta\mathbf{v}_j$ is an independent, zero-mean random vector with $\mathrm{Cov}(\delta\mathbf{v}_j) = \sigma_v^2\, \mathbf{I}$. Then for an arbitrary coordinate $c \in \{1, \ldots, d\}$, since the noises are independent, we have

$$
\begin{aligned}
\mathrm{Var}\big([\mathbf{e}^{(v)}]_c\big) &= \mathrm{Var}\Big(\sum_{j\in\mathcal{N}} \alpha_j\, \big[\delta\mathbf{v}_j\big]_c\Big) \\
&= \sum_{j\in\mathcal{N}} \alpha_j^2\, \mathrm{Var}\big([\delta\mathbf{v}_j]_c\big) \\
&= \sigma_v^2 \sum_{j\in\mathcal{N}} \alpha_j^2.
\end{aligned}
\tag{25}
$$

Since the attention weights $0 \le \alpha_j \le 1$ and there are exactly $m$ non-local tokens, we have

$$
\begin{aligned}
\mathrm{Var}\big([\mathbf{e}^{(v)}]_c\big) &\le \sigma_v^2\, m\, \max_{j\in\mathcal{N}} \alpha_j^2 = C_1\, \sigma_v^2, \\
&\text{with } C_1 := m\, \max_{j\in\mathcal{N}} \alpha_j^2.
\end{aligned}
\tag{26}
$$

*The constant $C_1$ is deterministic since it depends only on the current softmax weights. And every coordinate of the value-error term is bounded in variance by $C_1\sigma_v^2$.*

For the second key-propagated term, recall the first-order perturbation of each softmax weight

$$
\delta\alpha_j = \alpha_j\Big(e_j^{(k)} - \sum_k \alpha_k\, e_k^{(k)}\Big), \quad e_j^{(k)} := \frac{\mathbf{q}^\top \delta\mathbf{k}_j}{\sqrt{d}},
\tag{27}
$$

where the key-noise scalars $e_k^{(k)}$ are i.i.d., zero-mean with variance $\sigma_k^2$. Then for one output coordinate $c$, we need the variance of $\big[\mathbf{e}^{(k)}\big]_c = \sum_{j\in\mathcal{N}} \delta\alpha_j\, v_{j,c}$.

First, since $\delta\alpha_j$ is a linear combination of independent noises,

$$
\mathrm{Var}[\delta\alpha_j] = \alpha_j^2\, \sigma_k^2\Big(1 + \sum_k \alpha_k^2\Big) \le 2\, \alpha_j^2 \sigma_k^2,
\tag{28}
$$

because $\sum_k \alpha_k^2 \le 1$.

Then since each addend $\delta\alpha_j\, v_{j,c}$ is zero-mean, we have

$$
\begin{aligned}
\mathrm{Var}\big([\mathbf{e}^{(k)}]_c\big) &= \sum_{j\in\mathcal{N}} v_{j,c}^2\, \mathrm{Var}[\delta\alpha_j] \\
&\le 2\, \sigma_k^2\big(\max_{j\in\mathcal{N}} v_{j,c}^2\big) \sum_{j\in\mathcal{N}} \alpha_j^2.
\end{aligned}
\tag{29}
$$

With $\sum_{j\in\mathcal{N}} \alpha_j^2 \le m \max_j \alpha_j^2$,

$$
\begin{aligned}
\mathrm{Var}\big([\mathbf{e}^{(k)}]_c\big) &\le 2\, \sigma_k^2\, m\, \big(\max_{j\in\mathcal{N}} v_{j,c}^2\big)\big(\max_{j\in\mathcal{N}} \alpha_j^2\big) \\
&= C_2\, \sigma_k^2,
\end{aligned}
\tag{30}
$$

where

$$C_2 := 2\,m\,\max_{j \in \mathcal{N}}\big(\alpha_j^2 v_{j,c}^2\big). \tag{31}$$

*The constant $C_2$ is deterministic since it depends only on the current softmax weights and the fixed value vectors, not on the random noise. And every coordinate of the key-propagated error is bounded in variance by $C_2\sigma_k^2$.*

In conclusion, we prove that the mixed-precision attention output error $\boldsymbol{\delta}$ decomposes into a value-error term and a key-propagated term, and that each coordinate satisfies

$$\mathrm{Var}([\boldsymbol{\delta}]_c) \leq C_1\sigma_v^2 + C_2\sigma_k^2, \tag{32}$$

where $C_1, C_2$ are deterministic constants depending only on model parameters. Since $\sigma_k^2, \sigma_v^2$ are bounded in Equation 20, the attention computation error is properly bounded and decreases with larger codebook size $K$.

**Single Class Token Attention Output Error.** Its attention output error vector has $m = \frac{N-1}{N}T$ independent coordinates (i.e., the number of non-local tokens), each with per-coordinate variance $\sigma^2 := \mathrm{Var}\big([\mathbf{e}^{(v)}]_c\big) + \mathrm{Var}\big([\mathbf{e}^{(k)}]_c\big)$. Hence

$$\mathbb{E}\big[\|\boldsymbol{\delta}_{\mathrm{single}}\|_2^2\big] = m\,d\,\sigma^2. \tag{33}$$

**Distributed Class Tokens Attention Output Error.** Let $\boldsymbol{\delta}^{(i)}$ be the error vector for the $i$-th device. Vectors $\boldsymbol{\delta}^{(i)}$ are independent and identically distributed. The averaged class token output is $\bar{\boldsymbol{\delta}} = \frac{1}{N}\sum_{i=1}^{N}\boldsymbol{\delta}^{(i)}$, then we have

$$\mathbb{E}\big[\|\bar{\boldsymbol{\delta}}\|_2^2\big] = \frac{1}{N^2}\sum_{i=1}^{N}\mathbb{E}\big[\|\boldsymbol{\delta}^{(i)}\|_2^2\big] = \frac{1}{N}\,m\,d\,\sigma^2. \tag{34}$$

**Attention Error Comparison.** According to Equations 33 and 34, using distributed class tokens yields a factor of $1/N$ in the expected attention output error compared to the single class token, indicating a more accurate attention computation under the mixed-precision attention mechanism. $\qquad\square$

## D    RESULTS ON ACCURACY CONT.

**Experiments with Different Random Seeds.** To evaluate the robustness of ASTRA to randomness, we repeat each experiment ten times using different random seeds (0–9) on ImageNet-1K. As shown in Table 8, ASTRA consistently achieves stable performance across all group configurations, with standard deviations below 0.0012. These results demonstrate that ASTRA produces reproducible outcomes and is not sensitive to randomness in training.

Table 8: Accuracy of ASTRA on ImageNet-1K under ten runs with different random seeds (0–9). Mean and standard deviation (Std.) are reported for each group configuration. The original ViT-Base achieves 80.32% accuracy for reference.

| Seeds 0-9 | Mean | Std. |
|---|---|---|
| #Groups = 1 | 0.7681 | 0.0012 |
| #Groups = 16 | 0.7855 | 0.0008 |
| #Groups = 32 | 0.8002 | 0.0008 |

**Accuracy in Heterogeneous Settings.** In the main paper, we assume the computational workload is evenly distributed across homogeneous devices. To better evaluate the scalability of ASTRA in practical scenarios, we further explore its performance when deployed on heterogeneous devices. This section focuses on how heterogeneous deployment affects accuracy. Latency measurements are conducted on homogeneous devices to ensure consistency and are reported in the main paper and Appendix E.

In heterogeneous settings, stronger devices are assigned more tokens to balance the overall computation workload, while weaker devices receive fewer. Let $N$ be the total number of tokens and $K$ the

number of devices. Denote $n_k$ as the number of tokens on device $k$, such that $\sum_{k=1}^{K} n_k = N$. We define the *Full Precision Attention Rate (FPAR)* as:

$$\text{FPAR} = \sum_{k=1}^{K} \frac{n_k^2}{N^2}, \tag{35}$$

which measures the proportion of full-precision attention computation in the Mixed-Precision Attention mechanism. A higher FPAR indicates that more attention computation uses full-precision keys and values, thus better approximating standard attention.

To understand how FPAR captures token distribution imbalance, we examine its connection to the variance of $n_k$, which directly reflects distribution heterogeneity. Let $\mu = N/K$ be the average token count per device. Then:

$$\begin{aligned}
\text{Var}(n_k) &= \frac{1}{K} \sum_{k=1}^{K} (n_k - \mu)^2 \\
&= \frac{1}{K} \sum_{k=1}^{K} n_k^2 - \mu^2 \\
&= \frac{N^2}{K} \cdot (\text{FPAR} - \frac{1}{K}).
\end{aligned} \tag{36}$$

This shows that FPAR is a monotonic function of the variance of token allocation. In other words, as the load distribution becomes more imbalanced (i.e., more heterogeneous), FPAR increases.

To study how FPAR relates to model performance, we train ASTRA on ImageNet using #Groups $= 32$ across 4 devices. During training, tokens are randomly distributed across devices in each batch to simulate workload balancing on heterogeneous hardware. During evaluation, we continue to randomly assign tokens to devices and record both the prediction accuracy and the corresponding FPAR for each batch.

Figure 7 shows the distribution of FPAR across all evaluation batches. We divide the evaluation data into five bins based on FPAR, each containing 20% of the samples. Table 9 reports the mean accuracy for each bin. While the overall accuracy under heterogeneous deployment is slightly lower than in the homogeneous case—likely due to the added randomness in token assignment and the increased difficulty in learning a consistent pattern—we observe a clear positive correlation between FPAR and accuracy. This suggests that higher full-precision attention contributes to better model performance, demonstrating that our method remains effective under heterogeneous device settings.

Table 9: Accuracy of ASTRA under heterogeneous token distributions.

| FPAR Range | Mean Accuracy (%) |
|---|---|
| [0.2501, 0.2932] | 78.89 |
| [0.2932, 0.3196] | 78.96 |
| [0.3196, 0.3516] | 79.39 |
| [0.3516, 0.4020] | 79.62 |
| [0.4020, 0.7461] | 79.87 |

**Task Performance of Llama3-8B on Downstream Tasks.** We conducted additional experiments on four downstream sequence classification datasets, including CoLA Warstadt et al. (2019), SST2 Socher et al. (2013), AG News Zhang et al. (2015), and QQP Sharma et al. (2019). Table 10 reports the accuracy of the original Llama3-8B and its ASTRA versions. The results demonstrate that our small increases in pre-training perplexity in Table 6 lead to minor differences on downstream tasks, validating ASTRA 's capability in maintaining task performance.

# E    RESULTS ON INFERENCE LATENCY CONT.

In the main paper, we evaluate the inference latency of ASTRA under three key factors that impact multi-device performance: (1) **Inter-device bandwidth**, which affects the cost of communication

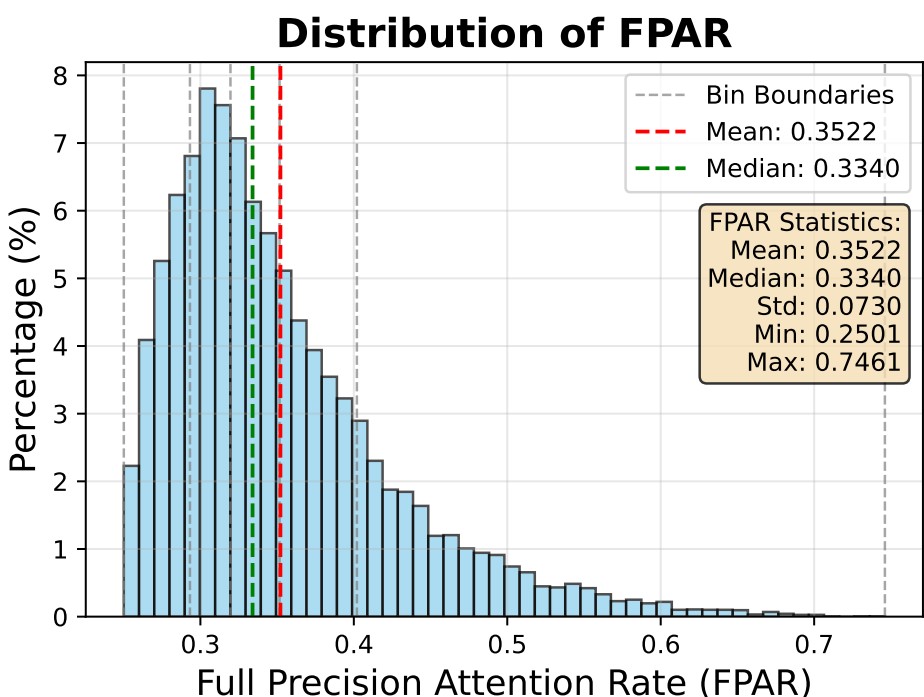

Figure 7: FPAR histogram across evaluation batches.

Table 10: Task performance of ASTRA on downstream tasks with Llama3-8B.

| Dataset | CoLA | SST2 | AG News | QQP |
|---------|------|------|---------|-----|
| Llama3-8B | 0.7615 | 0.8426 | 0.8374 | 0.7970 |
| ASTRA, G=1 | 0.7428 | 0.7545 | 0.7852 | 0.7703 |
| ASTRA, G=16 | 0.7451 | 0.8179 | 0.8292 | 0.7674 |
| ASTRA, G=32 | **0.7539** | **0.8314** | **0.8325** | **0.7803** |

across devices; (2) **Number of devices**, which determines the degree of parallelism; and (3) **Input token length**, which scales both computation and communication demands.

While the main text focuses on a representative configuration for each factor, this section presents additional results across a broader range of settings to further validate our findings. We include more granular evaluations varying each of the three dimensions to provide a comprehensive view of ASTRA's latency behavior under diverse deployment scenarios.

**Varying Bandwidth.** To assess the scalability of ASTRA under different communication constraints, we evaluate its end-to-end latency speedup across a wide range of inter-device bandwidths from 10 to 500 Mbps. Figure 8 presents the results across varying numbers of devices i.e., 2, 4, 6, 8) with a fixed input token length of 1024, while Figure 9 shows the results under varying input lengths (i.e., 256, 512, 1024, 2048, 4096) with 4 devices.

Across all configurations, ASTRA consistently outperforms existing multi-device baselines, with its advantage becoming increasingly significant as bandwidth decreases. For instance, in the 4-device and 1024-token setting (Figure 8(b)), ASTRA achieves a speedup of $2.64\times$ at 10 Mbps, while the strongest baseline (BP-SP, $N_b = 1$) only reaches $0.17\times$. This trend also holds for different token lengths. As shown in Figure 9, the relative speedup of ASTRA grows with increasing sequence length. These results confirm that ASTRA maintains high efficiency even under constrained bandwidth, thanks to its aggressive communication reduction strategy. In addition, grouped quantization (e.g., $G = 16, 32$) offers a tunable balance between compression and accuracy, enabling consistent latency benefits across a wide range of system settings.

**Varying Device Counts.** We further examine how the number of participating devices affects end-to-end latency speedup under varying bandwidth conditions. Figure 10 presents results for device counts ranging from 2 to 8 across bandwidths from 10 to 500 Mbps, with the input length fixed at 1024 tokens.

Across all bandwidth settings, ASTRA exhibits steadily increasing speedup as more devices are involved, demonstrating its ability to effectively utilize parallel computation. Thanks to its communication-efficient design, where non-local token embeddings are transmitted in compact vector-quantized form, ASTRA consistently delivers strong gains even under constrained bandwidth, highlighting its scalability across a range of deployment scenarios.

**Varying Input Token Length.** We evaluate how input token length affects latency speedup under different bandwidth conditions. Figure 11 shows results for input lengths from 256 to 4096 tokens across bandwidths from 10 to 500 Mbps using 4 devices. Across all bandwidth settings, ASTRA consistently achieves higher speedup than baseline methods and our improvements enhance as the input length increases. This trend highlights the communication bottleneck in existing methods, which becomes more significant with longer sequences, while ASTRA effectively mitigates this overhead through aggressive compression.

# F  ABLATION STUDY

**Varying Noise Magnitude $\lambda$.** The noise magnitude $\lambda$ in Noise-Augmented Vector Quantization controls the scale of noise added to the quantized embeddings during training. Table 11 shows the effect of varying $\lambda \in \{0.0, 0.1, 0.3, 1.0\}$ on both training and validation accuracy. All other hyperparameters are fixed (e.g., 16 groups, commitment loss weight $\beta = 0.0005$). As $\lambda$ increases, the gap between training and validation accuracy consistently decreases, indicating that injecting noise improves generalizability by preventing the model from overfitting to discrete embedding patterns. Notably, when $\lambda = 1.0$, the validation accuracy improves by 0.86% compared to $\lambda = 0$, demonstrating the effectiveness of our proposed strategy over naive vector quantization.

Table 11: The impact of noise magnitude $\lambda$ on classification accuracy. Gap = Train - Val.

| $\lambda$ | Train | Val | Gap |
|---|---|---|---|
| 0.0 | 99.98 | 89.91 | 10.07 |
| 0.1 | 99.97 | 90.02 | 9.95 |
| 0.3 | 99.98 | 90.13 | 9.85 |
| 1.0 | 99.98 | 90.77 | 9.21 |

**Distributed Class Token *VS* Single Class Token.** Table 12 reports the classification accuracy of ASTRA using either a single class token or distributed class tokens across devices. The distributed strategy consistently outperforms the single-token baseline, with accuracy gains ranging from 0.37% to 7.13% across different group configurations and commitment loss weights. This demonstrates that allowing class tokens to attend to all full-precision context tokens in a distributed manner significantly enhances their ability to aggregate global information.

Table 12: Distributed Class Token *VS* Single Class Token under different group configurations and commitment loss weights.

| $\beta$ | #Groups=1 | | | #Groups=16 | | | #Groups=32 | | |
|---|---|---|---|---|---|---|---|---|---|
| | Single | Dist. | $\triangle$ Acc. | Single | Dist. | $\triangle$ Acc. | Single | Dist. | $\triangle$ Acc. |
| 0.0001 | 82.39 | 88.95 | 6.56 | 89.11 | 90.37 | 1.26 | 90.39 | 91.60 | 1.21 |
| 0.0002 | 81.48 | 88.60 | 7.12 | 89.02 | 90.38 | 1.36 | 90.84 | 91.21 | 0.37 |
| 0.0005 | 81.82 | **88.95** | 7.13 | 88.93 | **90.77** | 1.84 | 90.79 | **91.64** | 0.85 |

**Varying the Commitment Loss Weights $\beta$.** Recall from Section 3.2 that we include a commitment loss term to encourage the original token embeddings to remain close to their assigned codebook entries following VQVAE (Van Den Oord et al., 2017). While the original VQVAE typically sets the commitment loss weight to 0.25, we adopt much smaller values in our setting, as we apply vector quantization separately at each Transformer block. Table 12 in the main paper reports the results of ASTRA under different commitment weights $\beta \in \{0.0001, 0.0002, 0.0005\}$. Here we further

compare with two control variants: one without commitment loss (i.e., $\beta = 0$), and one using an excessively large weight (i.e., $\beta = 0.25$). As shown in Table 13, either omitting or misconfiguring the commitment term slightly degrades accuracy, with performance drops ranging from 0.1% to 1.67%, confirming the importance of tuning $\beta$ appropriately.

Table 13: The impact of commitment loss weight $\beta$.

| $\beta$ | #Groups | | |
|---|---|---|---|
| | 1 | 16 | 32 |
| 0 | 88.85 | 90.46 | 91.42 |
| 0.25 | 88.75 | 89.7 | 89.97 |
| best | **88.95** | **90.77** | **91.64** |

## G  MEMORY COST ANALYSIS

ASTRA introduces a small additional memory cost to store the VQ codebooks, while the vector-quantized keys and values can reduce the memory required by the KV cache. We discuss these two aspects separately below. VQ codebook introduces a small additional memory cost. The memory footprint of the VQ codebooks is

$$M_{\text{codebook}} = L \cdot C \cdot K \cdot d \cdot b$$

where $L$ is the number of layers, $C$ is the number of codebooks per layer, $K$ is the codebook size (number of entries), $d$ is the hidden dimension, and $b$ is the number of bytes per value. Note that this expression is independent of the number of VQ groups. Grouped VQ partitions the hidden dimension into groups (i.e., $G$ groups of dimension $d/G$). Since $G \cdot (d/G) = d$, the total codebook size only scales with the full hidden dimension $d$, not with $G$. In practice, this overhead is small compared to the original model parameters. For example, in LLaMA-3-8B, we use $L = 32$, $C = 2$, $K = 1024$, $d = 1024$, $b = 2$ bytes (i.e., float16 precision). This gives

$$M_{\text{codebook}} = 32 \times 2 \times 1024 \times 1024 \times 2 \text{ bytes} = 134{,}217{,}728 \text{ bytes} = 128 \text{ MiB}. \tag{37}$$

Thus, for LLaMA-3-8B, the total VQ codebook storage is about 128 MiB, regardless of the number of VQ groups. This corresponds to roughly 0.78KV cache memory cost is reduced by VQed keys and values. ASTRA reduces KV cache memory by storing non-local keys and values as VQ indices instead of full-precision tensors. For an input sequence of length $N$, the KV cache memory of the original model is

$$M_{\text{KV}}^{\text{orig}} = 2 \cdot N \cdot L \cdot d \cdot b, \tag{38}$$

where the factor 2 accounts for keys and values, $L$ is the number of layers, $d$ is the hidden dimension, and $b$ is the number of bytes per value. With ASTRA, we assume $n_d$ devices, $G$ VQ groups, and an even partition of tokens across devices. Each device keeps its local tokens in full precision, while non-local tokens are cached as VQ indices (one index per group per token). The KV cache memory becomes

$$M_{\text{KV}}^{\text{ASTRA}} = 2\Big( \underbrace{\frac{N}{n_d} \cdot L \cdot d \cdot b}_{\text{local full-precision KV}} + \underbrace{(n_d - 1) \cdot \frac{N}{n_d} \cdot L \cdot G \cdot \frac{\log_2 K}{8}}_{\text{non-local KV stored as indices}} \Big), \tag{39}$$

where $K$ is the codebook size and $\log_2 K$ is the number of bits per VQ index. For LLaMA-3-8B, we use $N = 1024$, $L = 32$, $d = 1024$, $b = 2$ bytes, $n_d = 4$, $G = 32$, and $K = 1024$ (i.e., ($\log_2 K = 10$), so we have

$$M_{\text{KV}}^{\text{orig}} = 2 \cdot 1024 \cdot 32 \cdot 1024 \cdot 2 = 134{,}217{,}728 \text{ bytes} \approx 128 \text{ MiB}, \tag{40}$$

$$M_{\text{KV}}^{\text{ASTRA}} = 2\Big(\frac{1024}{4}\cdot 32\cdot 1024\cdot 2 + (4-1)\cdot\frac{1024}{4}\cdot 32\cdot 32\cdot\frac{10}{8}\Big) = 35{,}520{,}512 \text{ bytes} \approx 33.9 \text{ MiB}. \tag{41}$$

Thus, in this configuration, ASTRA uses only about 26.5% of the original KV cache memory.

## H   IMPACT STATEMENT AND LIMITATION

**Potential Societal Impact.** Our work aims to make large Transformer models more deployable in real-world environments by enabling efficient multi-device inference under limited bandwidth. This can expand the accessibility of powerful AI models to edge and consumer-grade devices, potentially benefiting applications in healthcare, accessibility, and low-connectivity regions. However, multi-device deployment may introduce new robustness and maintenance challenges. Unlike single-device inference, distributed inference requires reliable synchronization and communication among devices. Failures in individual devices or unstable connections can lead to degraded performance or unpredictable outputs. These issues can make such systems harder to debug, monitor, and guarantee correctness, especially in safety-critical applications.

**Limitation and Future Work.** While ASTRA achieves strong performance across vision and language tasks, we observe a degradation in zero-shot generalization in the GPT experiments (see Section 4.2). We hypothesize this is due to the limited expressiveness of the discrete embedding space introduced by vector quantization. Future work may explore hybrid compression strategies that retain generalization ability while still reducing communication costs. Additionally, our grouped vector quantization design requires maintaining a separate codebook for each group, which increases the overall storage footprint and may limit deployment flexibility across heterogeneous environments. As a future direction, we aim to investigate codebook sharing mechanisms or dynamically composable codebooks to reduce storage costs and enable bandwidth-aware adaptation without retraining.

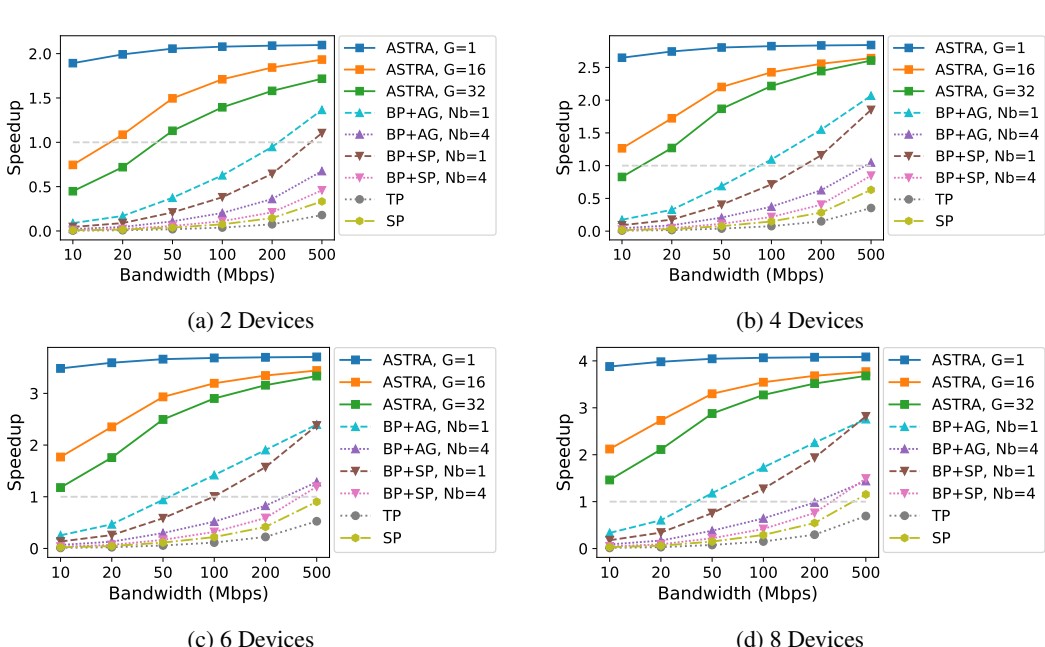

(a) 2 Devices

(b) 4 Devices

(c) 6 Devices

(d) 8 Devices

Figure 8: Speedup comparison under different bandwidth across different numbers of devices (w/ 1024 tokens).

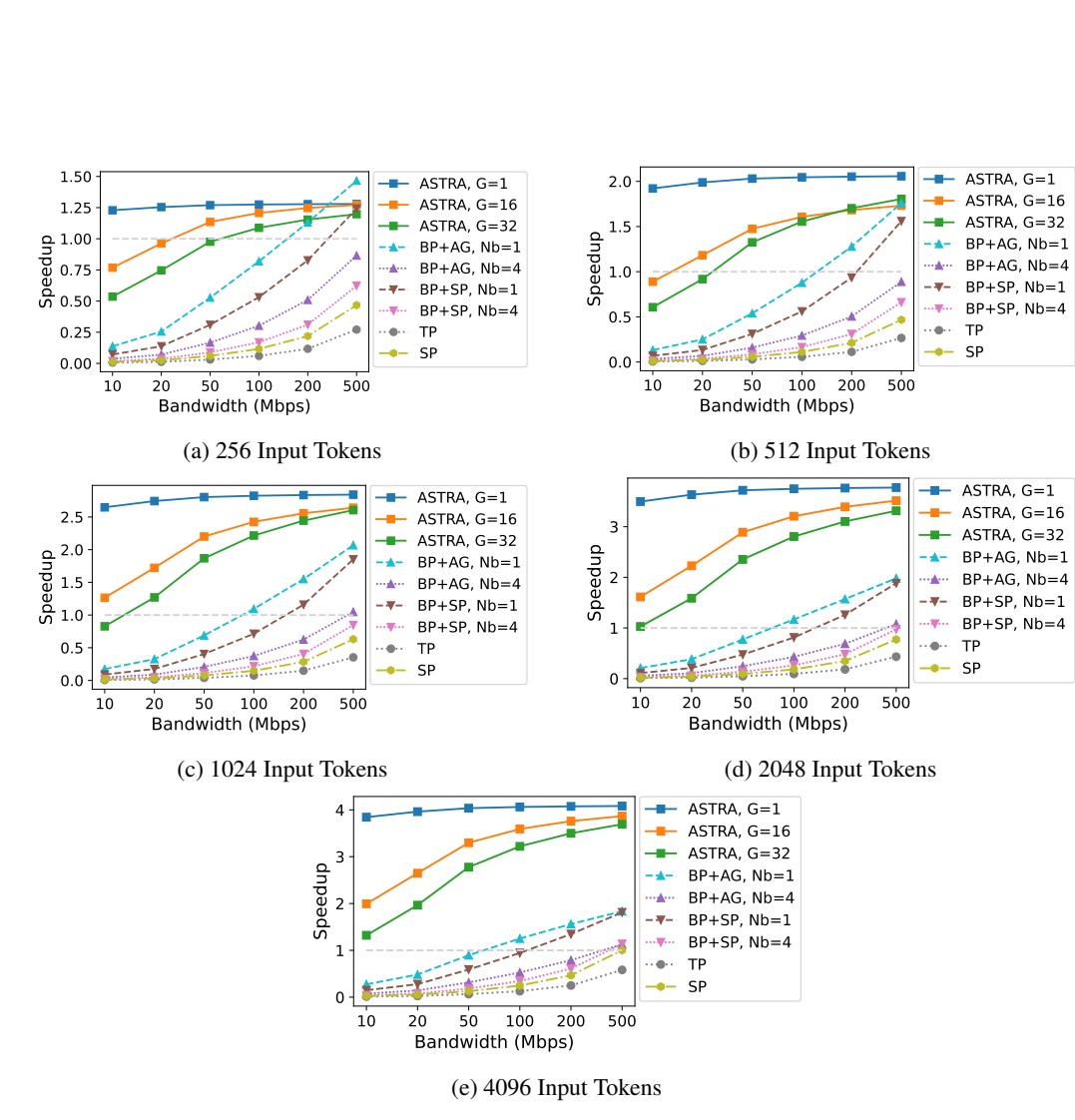

Figure 9: Speedup comparison under different bandwidth across different input token length (w/ 4 devices).

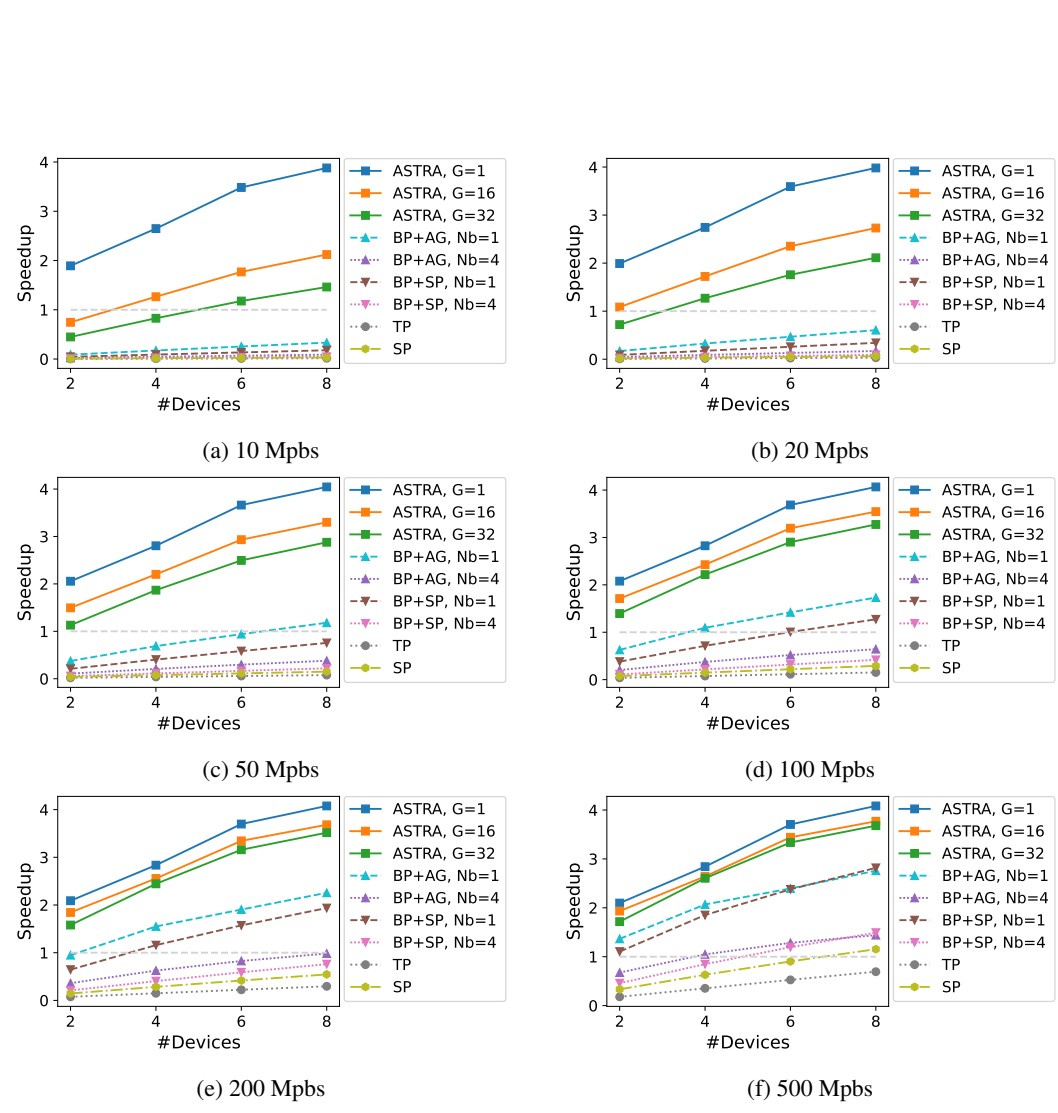

Figure 10: Speedup comparison under different devices across different bandwidth (w/ 1024 tokens).

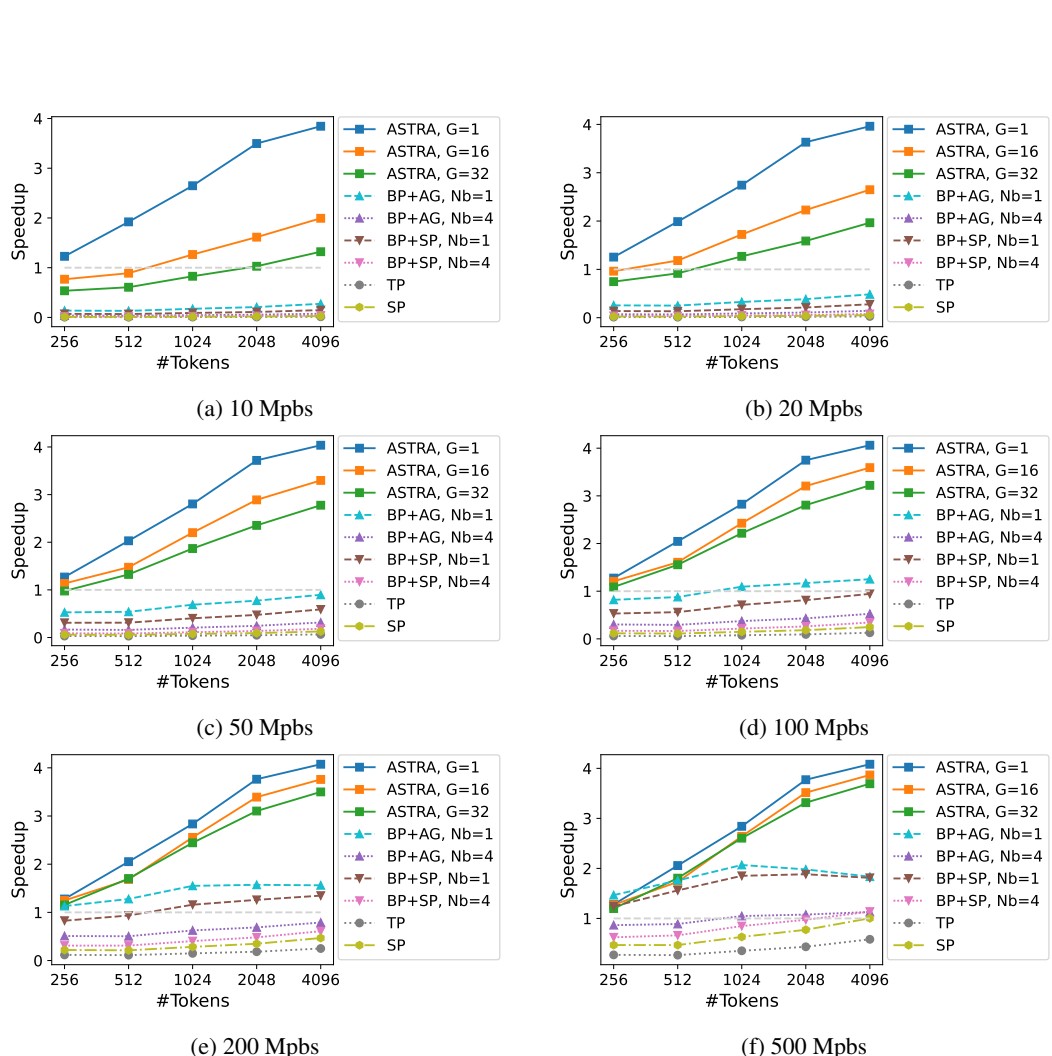

Figure 11: Speedup comparison under different input token length across different bandwidth (w/ 4 devices).

