# OpenReview forum: "Communication-Efficient Multi-Device Inference Acceleration for Transformer Models"
_ICLR.cc/2026/Conference — Submitted to ICLR 2026_

### Official Review · Reviewer_3hGB · 2025-10-17

**Soundness:** 2
**Presentation:** 3
**Contribution:** 2
**Rating:** 4
**Confidence:** 3

**Summary:**

This paper addresses the significant challenge of high inference latency in Transformer models, particularly in multi-device scenarios where inter-device communication becomes a dominant bottleneck in low-bandwidth environments. The authors propose ASTRA, a communication-efficient inference framework that builds on sequence parallelism but introduces a novel Mixed-Precision Attention mechanism to drastically reduce communication overhead. This mechanism computes attention using full-precision embeddings for local tokens while using low-bit vector-quantized (VQ) representations for non-local tokens transmitted between devices. To preserve model accuracy under such aggressive compression, ASTRA introduces two key optimizations: Noise-Augmented Quantization (NAVQ), a training-time regularization strategy that injects noise into quantized embeddings to improve generalization , and Distributed Class Tokens, which replicates the class token to each device and aggregates the outputs to mitigate information bias. Experiments on ViT and GPT-2 models show that ASTRA achieves substantial end-to-end speedups -- up to 2.64x over single-device inference and 15.25x over other multi-device methods -- in bandwidth-constrained settings (as low as 10 Mbps), while incurring only minor accuracy degradation.

**Strengths:**

The paper is pretty interesting wtih 2 proposed optimizations to speedup inference across multiple device. I am not a systems expert, but the math behind the optimizations is correct and makes sense.

**Weaknesses:**

The biggest weakness and drawback from the paper is the lack of large-scale experiments. Espeically the model choices in this paper is so small that none of these optimizations matter. Given the prevelance of open models of much larger sizes even ViTs and Qwen series models, it is empirically needed to make sure the proposed optimizations are quality neutral while providing the benefits. Without these results it hard to make a case for accepting the paper. I hope the authors can scale up the benchmarking further.

**Questions:**

see above.

---

> ### Author Response · Authors · 2025-11-22
> **Response to Reviewer 3hGB**
>
> Thank you for your interest and the suggestion regarding the large-scale experiments. Please refer to Q1 in *General Response to All Reviewers*, where we show the scalability of our method on a representative large transformer model, Llama-3-8B.
> For your convenience, we have pasted the detailed results and analysis below.
>
> ---
>
> We evaluate ASTRA on a large transformer model, **Llama-3-8B**, for next-token prediction on English Wikipedia dataset. 8-bit quantization is enabled for all methods, including the baselines and ASTRA, to execute inference with NVIDIA TitanX GPUs and keep fair comparisons. Results below show that **ASTRA scales to large models while maintaining performance close to the original and delivering substantial speedups under low bandwidth**.
>
> **Accuracy and Compression.** Table 1 reports perplexity (PPL; lower is better) together with communication overhead in bits per token as we vary the number of groups. Compared to the single-model baseline, **ASTRA maintains accuracy within a small margin while achieving a large communication reduction**. For example, at $G=1$, PPL increases from 5.81 (loss \= 1.76) to 8.06 (loss \= 2.08) for an over $1600\\times$ reduction in communicated datasize.
>
> **Table 1\.** Task accuracy and communication overhead on Wikipedia with Llama-3-8B. \#Groups is a hyperparameter in vector quantization.
>
> | Model | \#Groups | Bits per Token | Compression Ratio | PPL |
> | :---: | :---: | :---: | :---: | :---: |
> | Llama-3-8B | \- | 1,048,576 | \- | 5.8118 |
> | ASTRA | 1 | 640 | 1,638.4 | 8.0631 |
> | ASTRA | 16 | 10,240 | 102.4 | 7.9197 |
> | ASTRA | 32 | 20,480 | 51.2 | 7.5656 |
>
> **Latency under Low Bandwidth.** Table 2 summarizes end-to-end latency on 4 devices with 1024 tokens across bandwidths from 10–500 Mbps. ASTRA consistently outperforms state-of-the-art multi-device inference methods. For instance, under low bandwidth (e.g., 10–100 Mbps), **ASTRA attains $1.13-5.13\\times$ speedup over the fastest baseline, i.e., BP, Nb=4**. Because 8-bit quantization is applied to all methods, these gains isolate the contribution from ASTRA’s communication-efficient design.
>
> **Table 2\.** Latency (s) comparison between ASTRA and baselines on 4 devices with 1024 tokens across different bandwidths (Mbps). TP: Tensor Parallelism, SP: Sequence Parallelism, BP: Block Parallelism.
>
> | Bandwidth | 10 | 20 | 50 | 100 | 200 | 500 |
> | :---: | :---: | :---: | :---: | :---: | :---: | :---: |
> | Llama-3-8B | 4.578 |  |  |  |  |  |
> | TP | 430.952 | 216.291 | 87.449 | 44.499 | 23.025 | 10.140 |
> | SP | 28.256 | 14.939 | 6.888 | 4.215 | 2.857 | 2.052 |
> | BP, Nb=4 | 4.642 | 3.047 | 2.085 | 1.753 | 1.586 | **1.485** |
> | BP, Nb=8 | 8.011 | 4.780 | 2.773 | 2.101 | 1.762 | 1.561 |
> | ASTRA, G=1 | **1.563** | **1.549** | **1.547** | **1.545** | **1.541** | 1.540 |
> | ASTRA, G=16 | 1.661 | 1.659 | 1.595 | 1.572 | 1.559 | 1.548 |
> | ASTRA, G=32 | 1.940 | 1.796 | 1.661 | 1.630 | 1.603 | 1.583 |

---

> > ### Comment · Reviewer_3hGB · 2025-11-22
> >
> > Thanks for the response. While the comms cost reduction is impressive, the pplx drop is alarming and when you ladder it up the gap has a high potential of increasing as the initial points suggest.
> >
> > While I am not an expert (I will follow along other reviewer discussion and make an informed choice), this weakness in inference time accuracy drop for speedup is not practical making it hard to champion. I hope the authors will take the feedback and work on reducing the accuracy drop significantly with the high level takeaways being the same.

---

> > > ### Author Response · Authors · 2025-11-25
> > > **Response to Reviewer 3hGB**
> > >
> > > Thank you for the prompt follow-up. We understand the concern about accuracy degradation and would like to provide more explanation regarding the gap.
> > >
> > > We first show that the gap can be reduced via hyperparameter tuning. In our initial response, we use the default learning rate (3e-4) on Llama-3-8B, as in our other experiments. We find that lowering the learning rate to 1e-4 can reduce the perplexity gap, as shown in Table 1 below. (\#Groups=32 is still running, we’ll keep this table updated.)
> > >
> > > **Table 1\.** ASTRA’s task performance with different hyperparameters.
> > >
> > > | Model | \#Groups | Learning Rate | PPL | Loss |
> > > | :---: | :---: | :---: | :---: | :---: |
> > > | Llama-3-8B | \- | \- | 5.8118 | 1.759 |
> > > | ASTRA | 1 | 3e-4 | 8.0631 | 2.087 |
> > > | ASTRA | 1 | 1e-4 | **7.7336** | 2.045 |
> > > | ASTRA | 16 | 3e-4 | 7.9197 | 2.069 |
> > > | ASTRA | 16 | 1e-4 | **7.5879** | 2.026 |
> > >
> > > **The perplexity gap is well within accepted standards in the literature.**
> > > State-of-the-art quantization methods show comparable perplexity increases. Table 1 in Huang et al. (2024) evaluates quantization methods on LLaMA3-8B using WikiText2:
> > >
> > > * **FP16 baseline:** PPL \= 6.1
> > > * **4-bit RTN:** PPL \= 8.5 (\~39% increase)
> > > * **GPTQ:** PPL \= 7.0 (\~15% increase)
> > > * **AWQ:** PPL \= 7.1 (\~16% increase)
> > > * **SmoothQuant:** PPL \= 7.7 (\~26% increase)
> > >
> > > These methods demonstrate that 15-40% perplexity increases are accepted and deployed in production when they enable substantial efficiency gains. ASTRA’s degradation falls within this range.
> > >
> > > **Why the perplexity gap is acceptable.** The perplexity gap exaggerated the actual degradation in model quality. Since perplexity is defined as $\\mathrm{PPL} \= \\exp(\\mathcal{L})$, the difference corresponds to a much smaller increase in terms of loss. For example, as shown by our latest results in Table 1, at $G=1$, PPL from 5.81 to 7.73 corresponds to a loss increase roughly from 1.76 to 2.04, i.e., only a change of about 0.3.
> > >
> > > \[1\] Huang, Wei, et al. "An empirical study of llama3 quantization: From llms to mllms." Visual Intelligence 2.1 (2024): 36\.

---

> > > > ### Comment · Reviewer_3hGB · 2025-11-25
> > > >
> > > > Thanks a lot for the response. While I am not an expert on the systems side of things, I do have fair amount of experience with the training and performance of LLMs. Having said that, whenever we claim pplx drop is reasonable (or comparing it to a different axis of speedup) we ought to show the numbers of downstream to corroborate with the drops not affecting downstreams.
> > > >
> > > > re learning rate: it is great to see hparam tuning helps, but when presenting a strong algorithm it is imperative to show the right recipe to the best extent. I understand it is a costly process, but if there is a way to estimate this quickly it would be great.

---

> > > > > ### Author Response · Authors · 2025-12-03
> > > > > **Response to Reviewer 3hGB**
> > > > >
> > > > > Thank you for raising this important point. We conducted additional experiments on four downstream sequence classification datasets, including CoLA \[1\], SST2 \[2\], AG News \[3\], and QQP \[4\]. Table below reports the accuracy of the original Llama3-8B and its ASTRA versions. The results demonstrate that our small increases in pre-training perplexity lead to minor differences on downstream tasks, validating ASTRA’s capability in maintaining task performance.
> > > > >
> > > > > | Dataset | CoLA | SST2 | AG News | QQP |
> > > > > | :---: | :---: | :---: | :---: | :---: |
> > > > > | Llama3-8B | 0.7615 | 0.8426 | 0.8374 | 0.7970 |
> > > > > | ASTRA, G=1 | 0.7428 | 0.7545 | 0.7852 | 0.7703 |
> > > > > | ASTRA, G=16 | 0.7451 | 0.8179 | 0.8292 | 0.7674 |
> > > > > | ASTRA, G=32 | **0.7539** | **0.8314** | **0.8325** | **0.7803** |
> > > > >
> > > > > \[1\] CoLA:  Warstadt A, Singh A, Bowman S R. Neural network acceptability judgments. Transactions of the Association for Computational Linguistics, 2019\.
> > > > > \[2\] SST2: Socher R, Perelygin A, Wu J, et al. Recursive deep models for semantic compositionality over a sentiment treebank. EMNLP 2013\.
> > > > > \[3\] AG News: Zhang X, Zhao J, LeCun Y. Character-level convolutional networks for text classification. NeurIPS, 2015\.
> > > > > \[4\] QQP:  Sharma L, Graesser L, Nangia N, et al. Natural language understanding with the quora question pairs dataset. arXiv 2019\.
> > > > >
> > > > > Regarding the training recipe, we conducted an additional hyperparameter search over the learning rate and commitment-loss weight. Our best configuration uses a learning rate of 1e-4 and a commitment-loss weight of 1e-4, which narrows the perplexity gap to the base model. We have updated these improved results in Tables 1 and 3 in the general response to all the reviewers and Table 6 in the revised manuscript.

---

### Official Review · Reviewer_24hX · 2025-10-30

**Soundness:** 3
**Presentation:** 2
**Contribution:** 2
**Rating:** 4
**Confidence:** 4

**Summary:**

This paper introduces ASTRA, a communication-efficient framework for multi-device Transformer inference under bandwidth-constrained settings. Existing multi-device methods suffer from high inter-device communication overhead, which dominates latency when bandwidth is limited. ASTRA addresses this by combining sequence parallelism with a Mixed-Precision Attention mechanism: local attention is computed at full precision, while remote tokens are compressed via low-bit vector quantization.

**Strengths:**

This paper observes a real bottleneck in multi-device Transformer inference for low-bandwidth or edge environments, which is increasingly relevant for real-time AI applications.

**Weaknesses:**

1. ASTRA integrates known techniques (sequence parallelism + token quantization + noise augmentation), so the main contribution is in practical integration and bandwidth optimization, not a fundamentally new inference algorithm.
2. Lacks formal characterization of attention approximation error due to vector quantization and noise injection. Consider adding error bounds or theoretical analysis of how quantization and noise affect attention computation and model accuracy.
3. Experiments assume stable bandwidth; network variability or heterogeneous device scenarios are not tested.
4. Latency and energy claims are based on simulation; real-world multi-device deployment is not evaluated. Incorporate hardware-level profiling (multi-GPU, FPGA, or edge devices) to substantiate speedup and efficiency claims.

**Questions:**

see weakness

---

> ### Author Response · Authors · 2025-11-22
> **Response to Reviewer 24hX (1/2)**
>
> Thank you for your constructive comments and critical feedback. Below we provide one-to-one responses and detailed discussions.
>
> **W1:** ASTRA integrates known techniques (sequence parallelism \+ token quantization \+ noise augmentation), so the main contribution is in practical integration and bandwidth optimization, not a fundamentally new inference algorithm.
>
> **A1:** While ASTRA indeed builds on established concepts such as sequence parallelism and vector quantization, our contribution goes beyond a naive integration. ASTRA introduces three novel insights/designs to achieve communication-efficient multi-device Transformer inference with minimal accuracy loss.
>
> First, **ASTRA introduces a mixed-precision attention mechanism tailored for multi-device settings**. To the best of our knowledge, prior sequence-parallel or tensor-parallel methods usually keep all tokens in full precision for preserving accuracy. In contrast, ASTRA explicitly distinguishes local and remote tokens within the same attention layer: local tokens are kept in full precision, while only remote tokens are vector-quantized before communication. This asymmetric, token-wise mixed precision is designed specifically to reduce cross-device bandwidth without sacrificing local attention quality. We provide a theoretical analysis in Appendix C showing that, under bounded key and value quantization error, the resulting attention output error is rigorously bounded.
>
> Second, **we do not apply VQ in a trivial *compress-and-send* manner; instead we design a new noise-augmented VQ (NAVQ) scheme**. VQ is used as a tool to shrink the transmitted representations, but naive VQ introduces accuracy degradation. To address this, we propose noise-augmented VQ, which injects carefully calibrated noise based on the empirical VQ residuals. Both theoretically (Appendix B) and empirically (Appendix F), we show that NAVQ improves the distributional fidelity of the quantized embeddings to the original feature distribution, leading to higher task performance compared to standard VQ at the same bitrate. We are not aware of prior multi-device inference work that combines VQ with such a noise augmentation or provides corresponding theoretical analysis.
>
> Third, **ASTRA introduces distributed class tokens to further stabilize attention under quantization error**. Instead of relying on a single global class token, we maintain device-local class tokens and aggregate them to reduce the variance of the attention output under quantized remote tokens. This structural change is specifically designed to reduce the expected attention output error, and its benefit is validated through both our theory (Appendix C) and experiments (Appendix F).
>
> We explicitly highlight these contributions in the introduction, as they jointly enable a reliable and communication-efficient inference framework. We believe that, taken together, these innovations constitute a substantial and appropriate contribution to a top-tier venue.

---

> > ### Author Response · Authors · 2025-11-22
> > **Response to Reviewer 24hX (2/2)**
> >
> > **W2:** Consider adding error bounds or a theoretical analysis of how quantization and noise injection affect attention computation and model accuracy.
> >
> > **A2:** Our current theory in Appendix B and Appendix C already provides the requested error analysis.
> >
> > **How quantization affects attention computation.** In Appendix C, we prove that **the attention error is bounded under bounded quantization error.**
> > We first model the effect of quantization on keys and values as additive noise $\\delta k\_j, \\delta v\_j$ with zero mean and variances $\\sigma\_k^2, \\sigma\_v^2$. The two variances are bounded according to the classical high-rate VQ theory \[1,2\]. It shows that the mean-squared quantization error of an optimal $K$-level VQ in dimension $d$ satisfies
> > $\\mathbb{E}\\|\\mathbf{X} \- \\hat{\\mathbf{X}}\\|\_2^2 \\le C\_d \\cdot \\sigma\_X^2 \\cdot K^{-2/d},$
> > where $\\hat{\\mathbf{X}}$ denotes the quantized embedding of $\\mathbf{X}$, and $C\_d$ is a constant depending on the dimension $d$.
> > This implies a per-dimension variance bound
> > $\\sigma\_k^2, \\sigma\_v^2 \= \\frac{1}{d}\\mathbb{E}\\|\\mathbf{X} \- \\hat{\\mathbf{X}}\\|\_2^2 \\le \\tilde{C} \\cdot \\sigma\_X^2 \\cdot K^{-2/d}.$
> >
> > Then we prove that the mixed-precision attention output error $\\boldsymbol\\delta$ decomposes into a value-error term and a key-propagated term, and that each coordinate satisfies
> > $\\mathrm{Var}(\[\\boldsymbol\\delta\]\_c) \\le C\_1 \\sigma\_v^2 \+ C\_2 \\sigma\_k^2,$
> > where $C\_1, C\_2$ are deterministic constants depending only on model parameters.
> > Therefore, since the $\\sigma\_k^2, \\sigma\_v^2$ are bounded proved by high-rate quantization theory, the attention output error is properly bounded and decreases with larger codebook size $K$.
> >
> > **How noise injection affects attention computation.** In Appendix B, we prove that our noise-augmented VQ produces embeddings whose distribution is closer to the original features than standard VQ in Wasserstein-2 distance, implying a strictly smaller quantization and attention output error with the same boundedness.
> >
> > \[1\] P. L. Zador, “Asymptotic quantization error of continuous signals and the quantization dimension,” IEEE Transactions on Information Theory, vol. 28, no. 2, pp. 139–149, 1982\.
> > \[2\] R. M. Gray and A. Gersho, Vector Quantization and Signal Compression. Springer Science & Business Media, 1991\.
> >
> > ---
> >
> > **W3:** Network variability or heterogeneous device scenarios are not tested.
> >
> > **A3:** For network variability, please refer to Q2 in *General Response to All Reviewers*, where we provide additional experiments under both varied bandwidth and 5% random packet loss. Under these realistic network conditions, ASTRA maintained task performance and achieved higher inference throughput than single-device and multi-device baselines.
> >
> > For heterogeneous devices, Appendix D provides experiments where tokens are distributed in proportion to device capability so that stronger devices process more tokens. As heterogeneity increases, a larger fraction of attention computation uses full-precision keys and values on the stronger devices, which better approximates standard attention and yields higher accuracy. These results demonstrate that ASTRA remains effective and scalable under heterogeneous-device scenarios.
> >
> > ---
> >
> > **W4:** Latency claims are based on simulation; real-world multi-device deployment is not evaluated.
> >
> > **A4:** We appreciate the concern and agree that a real multi-device deployment would be valuable. Nevertheless, we believe our current evaluation adequately reflects ASTRA’s advantages over state-of-the-art methods.
> >
> > Our work targets distributed collaborative inference under limited bandwidth, where inter-device transfers dominate end-to-end latency. Accordingly, the paper evaluates methods at fixed, representative bandwidths (10–500 Mbps) to directly evaluate this bottleneck. In addition, following the reviewers’ suggestion, we augment the evaluation with realistic network variability by replaying a variable-bandwidth trace and introducing a 5% packet-loss setting (see Q2 in *General Response to All Reviewers*).
> >
> > In summary, while a full distributed deployment is promising future work, the combination of controlled fixed-bandwidth tests and added variability experiments shows that ASTRA’s latency gains arise from its advanced communication-efficient design and persist under practical network conditions.

---

### Official Review · Reviewer_mHin · 2025-11-01

**Soundness:** 3
**Presentation:** 3
**Contribution:** 3
**Rating:** 6
**Confidence:** 3

**Summary:**

This paper introduces ASTRA, a framework for accelerating transformer inference across multiple devices in bandwidth-constrained environments. The key part is a Mixed-Precision Attention mechanism that computes local attention with full precision while using vector-quantized embeddings for non-local tokens, reducing communication overhead. To preserve accuracy under aggressive compression, the authors propose Noise-Augmented Vector Quantization (NAVQ) and Distributed Class Tokens. Experiments on ViT and GPT-2 models across vision and NLP tasks demonstrate speedups of up to 2.64x over single-device inference and 15.25x over existing multi-device methods at bandwidths as low as 10 Mbps, while maintaining accuracy within 3.58% of the original models.

**Strengths:**

1. Addresses a real bottleneck: The paper identifies and tackles a genuine problem, that communication dominates latency (58.6-93.5%) in bandwidth-constrained multi-device inference.
2. Novel compression approach: The Mixed-Precision Attention mechanism is creative, using full-precision for local tokens and VQ for remote tokens.
3. Good evaluation: Extensive experiments across multiple architectures (ViT, GPT-2), tasks (classification, language modeling), and conditions (bandwidth, device count, heterogeneity) demonstrate broad applicability.
4. Practical compatibility: The framework integrates with existing quantization methods (8-bit, 4-bit), showing additional speedups of 1.35 to 2.73x when combined.

**Weaknesses:**

1. Limited architectural types: The evaluation focuses only on ViT and GPT-2, which are relatively small and dated models. Modern applications use much larger models (e.g., LLaMA variants). The scalability claims are weakened without evidence on contemporary, production-scale models.

2. Severe zero-shot degradation: Table 3 shows large performance drops in zero-shot settings (e.g., GPT-2M perplexity increases from 43.22 to 62.29, a 44% degradation). This is a critical limitation for practical deployment where generalization is essential.

3. Baselines potentially unfair: The comparison with BP, TP, and SP assumes these methods use full float32 precision. However, these methods could also be combined with standard compression techniques (gradient compression, activation compression). A more fair comparison would evaluate "BP+8bit quantization" vs "ASTRA+8bit quantization" to isolate ASTRA's contribution. Additionally, recent methods like FlexGen or DistServe are not compared, making it unclear how ASTRA compares to the current sota.

4. Communication model oversimplified: The paper assumes fixed bandwidth and doesn't account for real-world network variability, packet loss, or latency jitter. The latency model appears to assume perfect overlap of computation and communication, which is rarely achievable. Dynamic bandwidth fluctuations common in WiFi environments (cited as the target deployment) could significantly impact the practical speedups. The authors should evaluate under realistic network conditions with variable bandwidth and packet loss, or at minimum discuss how ASTRA degrades under non-ideal conditions.

**Questions:**

Please address the implicitly listed questions in the weakness.

Can you provide a decision tree or heuristic for practitioners to select:
1. Number of groups G based on task type (vision vs. NLP) and target accuracy?
2. Commitment loss weight ε based on model architecture and dataset?
3. When to use distributed vs. single class tokens?

---

> ### Author Response · Authors · 2025-11-22
> **Response regarding weaknesses to Reviewer mHin**
>
> Thank you for your positive and constructive comments\! Below we provide one-to-one responses to the weaknesses and questions.
>
> **W1:** The scalability claims are weakened without evidence on contemporary, production-scale models.
>
> **A1:** Please refer to Q1 in *General Response to All Reviewers*, where we show the scalability of our method on a representative large transformer model, Llama-3-8B. Thank you\!
>
> ---
>
> **W2:** Table 3 shows large performance drops in zero-shot settings.
>
> **A2:** We acknowledge the performance drop observed in the zero-shot evaluation as mentioned in the limitation part of Appendix G. This behavior is expected, as our framework reduces communication cost through vector quantization and quantization typically trades off accuracy for improved efficiency.
>
> We’d like to emphasize that the accuracy degradation is moderate relative to the achieved communication gain. Perplexity (PPL) is an exponential form of token-level cross-entropy loss $\\mathcal{L}$, i.e., PPL=$\\exp(\\mathcal{L})$, meaning that a seemingly large PPL increase corresponds to a small additive rise in $\\mathcal{L}$. For example, for GPT2-M in Table 3, the zero-shot PPL increases from 43.22 to 62.29, which corresponds to an increase of \~0.36 in loss per token. This small increase accompanies a $\>100\\times$ reduction in transmission.
>
> ---
>
> **W3:** The baselines, i.e., BP, TP, SP, could also be combined with standard compression techniques. A more fair comparison would evaluate "BP+8bit quantization" vs "ASTRA+8bit quantization" to isolate ASTRA's contribution.
>
> **A3:** We believe our comparison setup is fair and already isolates ASTRA’s contribution. In Section 4.3, all methods (BP, TP, SP, and ASTRA) are evaluated under the same model and FP32 precision to form fair comparisons. This setting removes the influence of single-device acceleration such as quantization, ensuring that the observed latency differences reflect ONLY the effect of how each method partitions computation and transmits activations. Under these identical conditions, ASTRA consistently achieves lower end-to-end latency than all baselines under low bandwidth.
>
> Following your suggestion, we also include additional comparison on Llama-3-8B with 8-bit quantization enabled for all methods in *General Response to All Reviewers*. As shown in Table 2 in Q1, **ASTRA \+ 8-bit outperforms BP \+ 8-bit by up to $5.13\\times$ at the low bandwidth.** The results highlight ASTRA’s advantage of its fundamentally communication-efficient design.
>
> ---
>
> **W4:** Recent methods like FlexGen or DistServe are not compared, making it unclear how ASTRA compares to the current sota.
>
> **A4:** Our work targets distributed collaborative inference under limited bandwidth, so we compare against the state of the art in this direction. Specifically, we benchmark the strongest multi-device baselines in each family, i.e., Voltage \[1\] for SP and DeTransformer \[2\] for BP. Both are recent works published in top venues. FlexGen \[3\] and DistServe \[4\] are not suitable or strong baselines in our case, since FlexGen is a single-GPU system and DistServe is designed for high-bandwidth, low-latency interconnects (e.g., NVLink).
>
> \[1\] Hu C, Li B. When the edge meets transformers: Distributed inference with transformer models. 2024 IEEE 44th International Conference on Distributed Computing Systems (ICDCS). IEEE, 2024: 82-92.
> \[2\] Du J, Wei Y, Ye S, et al. Co-designing transformer architectures for distributed inference with low communication. IEEE Transactions on Parallel and Distributed Systems, 2024\.
> \[3\] Sheng Y, Zheng L, Yuan B, et al. Flexgen: High-throughput generative inference of large language models with a single gpu. International Conference on Machine Learning. PMLR, 2023: 31094-31116.
> \[4\] Zhong Y, Liu S, Chen J, et al. {DistServe}: Disaggregating prefill and decoding for goodput-optimized large language model serving. 18th USENIX Symposium on Operating Systems Design and Implementation (OSDI 24). 2024: 193-210.
>
> ---
>
> **W5:** The authors should evaluate under realistic network conditions with variable bandwidth and packet loss, or at minimum, discuss how ASTRA degrades under non-ideal conditions.
>
> **A5:** Please refer to Q2 in *General Response to All Reviewers*, where we provide additional experiments for the task performance and the inference request throughput under non-ideal network conditions. Thank you\!

---

> > ### Author Response · Authors · 2025-11-22
> > **Response regarding questions to Reviewer mHin**
> >
> > **Question:** Can you provide a heuristic for practitioners to select:
> > 1\. Number of groups G based on task type (vision vs. NLP) and target accuracy?
> > 2\. Commitment loss weight based on model architecture and dataset?
> > 3\. When to use distributed vs. single class tokens?
> >
> > **Answer:**
> > We agree that practitioners would benefit from simple rules of thumb. **Our guiding principle is to balance accuracy and latency under a given bandwidth budget**, and we observe the same trend across vision and NLP tasks.
> >
> > **Choosing the number of groups $G$.** There is an inherent trade-off: larger $G$ preserves accuracy but limits speedup, while smaller $G$ yields higher speedup at some accuracy cost. As indicated in Figure 10 of Appendix E, we recommend **adapting $G$ according to the available bandwidth**. When bandwidth is at least 100 Mbps, $G=32$ retains accuracy close to the baseline while improving latency. When bandwidth is around 10–20 Mbps, $G=1$ is preferable to prioritize latency.
> >
> > **Setting the commitment-loss weight.** We experimented with varying commitment loss weight and **empirically suggested using 0.0001-0.0005** as shown in Appendix F Table 11\. Omitting or mis-tuning this term may yield minor accuracy drops, i.e., 0.1%-1.67% shown in Appendix F Table 12\.
> >
> > **Distributed vs. single class tokens.** Appendix F Table 11 shows that distributed class tokens consistently outperform a single class token, with gains of 0.37%–7.13%. We therefore **recommend using distributed class tokens by default**, and resorting to a single token only in extremely memory-constrained deployments where a small accuracy loss is acceptable.

---

> > > ### Author Response · Authors · 2025-11-27
> > > **A Kind Reminder of Follow-up on Our Rebuttal**
> > >
> > > Hello Reviewer mHin,
> > >
> > > Many thanks for your review and for considering our rebuttal. If there are any remaining concerns or points that we could clarify, please feel free to let us know.

---

### Official Review · Reviewer_7h8e · 2025-11-03

**Soundness:** 3
**Presentation:** 4
**Contribution:** 3
**Rating:** 6
**Confidence:** 2

**Summary:**

The paper introduces a way to improve communication efficiency for multi device Transformer inference using sequence parallelism and mixed-precision. They show significant speedups on ViT and GPT2 scale models.

**Strengths:**

- The paper is well-presented and easy to follow
- Communication overhead is significant in large Transformer model distributed settings
- The use of codebooks is interesting

**Weaknesses:**

- The models used for inference are small and it is not clear to me that these hold at scale.

**Questions:**

- It would be helpful to further motivate the wireless and edge deployment motivation for this framework. Why are we doing inference requests on wifi?

---

> ### Author Response · Authors · 2025-11-22
> **Response to Reviewer 7h8e**
>
> Thank you for your positive comments\!  Below we provide one-to-one responses to the weakness and question.
>
> **Weakness:** The models used for inference are small and it is not clear to me that these hold at scale.
>
> **Response:** Please refer to Q1 in *General Response to All Reviewers*, where we show the scalability of our method on a representative large transformer model, Llama-3-8B. Thank you\!
>
> ---
>
> **Question:** It would be helpful to further motivate the wireless and edge deployment motivation for this framework. Why are we doing inference requests on wifi?
>
> **Response:** Our work targets deep learning inference on edge devices because edge devices (e.g., local computers) are closer to data source and utilizing these devices can provide a more private, responsive, and economical alternative compared to cloud-based inference. Individual edge devices often lack sufficient computational resources to run large models quickly. By distributing the inference workload across multiple devices collaboratively, we can achieve faster inference times than any single device could provide alone, making large model inference practical in edge environments.
>
> In typical home or local settings, edge devices are often connected through networks whose bandwidth is far lower than data-center links. Our method is therefore designed to maintain effectiveness under such constraints. The proposed framework is transport-agnostic, and its advantages apply equally to wireless connections such as Wi-Fi  (50-300 Mbps) and wired Ethernet (100 Mbps-1 Gbps).

---

> > ### Author Response · Authors · 2025-11-27
> > **A Kind Reminder of Follow-up on Our Rebuttal**
> >
> > Hello Reviewer 7h8e,
> >
> > Many thanks for your review and for considering our rebuttal. If there are any remaining concerns or points that we could clarify, please feel free to let us know.

---

### Author Response · Authors · 2025-11-22
**General Response to All Reviewers (1/2)**

Thank you for your comments and suggestions. We first provide general responses to all reviewers’ common questions. **The manuscript is also revised to include the additional experiments in Section 4.5**.

**Q1: Can the proposed method, ASTRA, scale to large Transformer Models? (Reviewer 7h8e, mHin, 3hGB)**

**A1:** We additionally evaluate ASTRA on a large transformer model, **Llama-3-8B**, for next-token prediction on English Wikipedia dataset \[1\]. 8-bit quantization is enabled for all methods, including the baselines and ASTRA, to execute inference with NVIDIA TitanX GPUs and keep fair comparisons. Results below show that **ASTRA scales to large models while maintaining performance close to the original and delivering substantial speedups under low bandwidth**.

**Accuracy and Compression.** Table 1 reports perplexity (PPL; lower is better) together with communication overhead in bits per token as we vary the number of groups. Compared to the single-model baseline, **ASTRA maintains accuracy within a small margin while achieving a large communication reduction**. For example, at $G=1$, PPL increases from 5.81 (loss \= 1.76) to 7.73 (loss \= 2.04) for an over $1600\\times$ reduction in communicated datasize.

**Table 1\.** Task accuracy and communication overhead on Wikipedia with Llama-3-8B. \#Groups is a hyperparameter in vector quantization.

| Model | \#Groups | Bits per Token | Compression Ratio | PPL |
| :---: | :---: | :---: | :---: | :---: |
| Llama-3-8B | \- | 1,048,576 | \- | 5.8118 |
| ASTRA | 1 | 640 | 1,638.4 | 7.7336 |
| ASTRA | 16 | 10,240 | 102.4 | 7.5879 |
| ASTRA | 32 | 20,480 | 51.2 | 7.4360 |

**Latency under Low Bandwidth.** Table 2 summarizes end-to-end latency on 4 devices with 1024 tokens across bandwidths from 10-500 Mbps. ASTRA consistently outperforms state-of-the-art multi-device inference methods. For instance, under low bandwidth (e.g., 10–100 Mbps), **ASTRA attains $1.13-5.13\\times$ speedup over the fastest baseline, i.e., BP, Nb=4**. Because 8-bit quantization is applied to all methods, these gains isolate the contribution from ASTRA’s communication-efficient design.

**Table 2\.** Latency (s) comparison between ASTRA and baselines on 4 devices with 1024 tokens across different bandwidths (Mbps). TP: Tensor Parallelism, SP: Sequence Parallelism, BP: Block Parallelism.

| Bandwidth | 10 | 20 | 50 | 100 | 200 | 500 |
| :---: | :---: | :---: | :---: | :---: | :---: | :---: |
| Llama-3-8B | 4.578 |  |  |  |  |  |
| TP | 430.952 | 216.291 | 87.449 | 44.499 | 23.025 | 10.140 |
| SP | 28.256 | 14.939 | 6.888 | 4.215 | 2.857 | 2.052 |
| BP, Nb=4 | 4.642 | 3.047 | 2.085 | 1.753 | 1.586 | **1.485** |
| BP, Nb=8 | 8.011 | 4.780 | 2.773 | 2.101 | 1.762 | 1.561 |
| ASTRA, G=1 | **1.563** | **1.549** | **1.547** | **1.545** | **1.541** | 1.540 |
| ASTRA, G=16 | 1.661 | 1.659 | 1.595 | 1.572 | 1.559 | 1.548 |
| ASTRA, G=32 | 1.940 | 1.796 | 1.661 | 1.630 | 1.603 | 1.583 |

\[1\] Wikimedia Foundation, https://dumps.wikimedia.org

---

> ### Author Response · Authors · 2025-11-22
> **General Response to All Reviewers (2/2)**
>
> **Q2: How does ASTRA perform under realistic network conditions with variable bandwidth and packet loss? (Reviewer mHin, 24hX)**
>
> **A2**: We further evaluate Llama-3-8B while stressing the inter-device communications with (i) packet loss and (ii) time-varying bandwidth. Results show that ASTRA maintains task accuracy under modest packet loss and delivers higher throughput than single-device and multi-device baselines when bandwidth fluctuates.
>
> **5% Packet Loss without Retransmission.** Since WiFi networks generally experience packet loss rates of around 1% to 5% \[1\], Table 3 reports perplexity when we apply a 5% random packet loss rate and do not enable any resend or repair. ASTRA preserves task performance under this packet loss. For example, PPL marginally increases from 7.43 to 7.44 at $G=32$. For $G=1$, the PPL doesn’t increase or even slightly decrease, likely due to randomness. This robustness arises because ASTRA communicates compact, grouped representations and introduces mixed-precision attention computation, reducing sensitivity to occasional missing packets from other devices.
>
> **Table 3\.** Task accuracy on Wikipedia with Llama-3-8B under a packet loss rate of 5%.
>
> | Model | \#Groups | PPL without packet loss | PPL with 5% packet loss |
> | :---: | :---: | :---: | :---: |
> | ASTRA | 1 | 7.7336 | 7.7294 |
> | ASTRA | 16 | 7.5879 | 7.5900 |
> | ASTRA | 32 | 7.4360 | 7.4431 |
>
> **Variable Bandwidth with 600-s Trace Replay.** Table 4 summarizes inference request throughput on 4 devices with 1024 tokens and a single batch size when the communication capacity varies over a fixed 600-second trace. To ensure realism, we simulate fluctuating network conditions using bandwidth traces generated by a Markovian model from Pensieve \[2\], where each state corresponds to an average throughput between 20-100 Mbps and transitions are biased toward nearby states to capture temporal correlation. We include the network trace in Figure 6 of Section 4.5 in the revised paper. Under this trace, ASTRA attains higher end-to-end throughput compared to both single-device inference and other multi-device inference methods.
>
> **Table 4\.** Inference request throughput (\#request/second) on 4 devices with 1024 tokens under varied network bandwidth over a fixed 600-second trace.
>
> | Model | Llama-3-8B Single Device | TP | SP | BP, Nb=4 | BP, Nb=8 | ASTRA, G=1 | ASTRA, G=16 | ASTRA, G=32 |
> | :---: | :---: | :---: | :---: | :---: | :---: | :---: | :---: | :---: |
> | Throughput | 0.218 | 0.01 | 0.141 | 0.493 | 0.37 | **0.653** | **0.625** | **0.607** |
>
> \[1\] Sheshadri R K, Koutsonikolas D. On packet loss rates in modern 802.11 networks. IEEE INFOCOM 2017-IEEE Conference on Computer Communications. IEEE, 2017: 1-9.
> \[2\] Mao H, Netravali R, Alizadeh M. Neural adaptive video streaming with Pensieve. Proceedings of the conference of the ACM special interest group on data communication. 2017: 197-210.

---

### Comment · Reviewer_24hX · 2025-11-26

Thanks a lot for the response.  However, my primary concerns remain unaddressed. After carefully considering the rebuttal and other reviewers’ comments, I will maintain my original score.

---

> ### Author Response · Authors · 2025-11-27
>
> Thank you again for the time and effort you have put into reviewing our paper and for your comment.
>
> We understand that you decided to keep your original score, and we fully respect that. For further improving this work, would you mind to clarify which of your primary concerns remain unaddressed?
>
> We truly appreciate your time and any additional guidance you provide.

---

> ### Comment · Reviewer_24hX · 2025-11-28
>
> 1) Your method uses information from different precision levels. However, the paper does not explain the cost of doing this. In most cases, keeping multiple precision copies can improve speed, but it also increases storage use. I did not find a discussion or report of this overhead. Adding this information would make the work clearer.
> 2) It is not clear whether your method works with ring attention in a distributed setup. The paper does not explain how the design fits such a scenario.
> If you can describe this part in more detail, I will consider giving a higher score.

---

> > ### Author Response · Authors · 2025-12-03
> > **Response to Review 24hX**
> >
> > Thank you for your follow-up questions. We provide the detailed analysis for each question below.
> >
> > **Q1: About detailed memory usage.**
> >
> > ASTRA introduces a small additional memory cost to store the VQ codebooks, while the vector-quantized keys and values can *reduce* the memory required by the KV cache. We discuss these two aspects separately below.
> >
> > **VQ codebook introduces a small additional memory cost.**
> >
> > The memory footprint of the VQ codebooks is $M\_{\\text{codebook}} \= L \\cdot C \\cdot K \\cdot d \\cdot b$, where $L$ is the number of layers, $C$ is the number of codebooks per layer, $K$ is the codebook size (number of entries), $d$ is the hidden dimension, and $b$ is the number of bytes per value.
> >
> > Note that this expression is independent of the number of VQ groups. Grouped VQ partitions the hidden dimension into groups (i.e., $G$ groups of dimension $d / G$). Since $G \\cdot (d / G) \= d$, the total codebook size only scales with the full hidden dimension $d$, not with $G$.
> >
> > In practice, this overhead is small compared to the original model parameters. For example, in LLaMA-3-8B, we use $L \= 32$, $C \= 2$, $K \= 1024$, $d \= 1024$, $b \= 2$ bytes (i.e., float16 precision). This gives
> >
> > $$
> >  M\_{\\text{codebook}}
> >  \= 32 \\times 2 \\times 1024 \\times 1024 \\times 2 \\text{ bytes}
> >  \= 134{,}217{,}728 \\text{ bytes} \= 128 \\text{ MiB}.
> >  $$
> >
> > Thus, for LLaMA-3-8B, the total VQ codebook storage is about 128 MiB, regardless of the number of VQ groups. This corresponds to roughly **0.78%** of the model size when the base model is in float16 ($\\approx16$ GB), and about **1.56%** when the base model is 8-bit quantized ($\\approx8$ GB).
> >
> > **ASTRA reduces KV cache memory cost.**
> >
> > ASTRA reduces KV cache memory by storing **non-local** keys and values as VQ indices instead of full-precision tensors. For an input sequence of length $N$, the KV cache memory of the original model is $M_{\text{KV}}^{\text{orig}} = 2 \cdot N \cdot L \cdot d \cdot b$, where the factor 2 accounts for keys and values, $L$ is the number of layers, $d$ is the hidden dimension, and $b$ is the number of bytes per value.
> >
> > With ASTRA, we assume $n\_d$ devices, $G$ VQ groups, and an even partition of tokens across devices. Each device keeps its **local** tokens in full precision, while **non-local** tokens are cached as VQ indices (one index per group per token). The KV cache memory becomes
> >
> > $$
> > M_{\text{KV}}^{\text{ASTRA}}
> > = 2 \Big(
> > \underbrace{\frac{N}{n_d} \cdot L \cdot d \cdot b}_{\text{local full-precision KV}}
> > \+
> > $$
> >
> > $$
> > \underbrace{(n_d - 1) \cdot \frac{N}{n_d} \cdot L \cdot G \cdot \frac{\log_2 K}{8}}_{\text{non-local KV stored as indices}}
> > \Big),
> > $$
> >
> > where $K$ is the codebook size and $\\log\_2 K$ is the number of bits per VQ index.
> >
> > For LLaMA-3-8B, we use $N \= 1024, L \= 32, d \= 1024, b \= 2$ bytes, $n\_d \= 4$, $G \= 32$, and $K \= 1024$ (i.e., ($\\log\_2 K \= 10$), so we have
> >
> > $$
> >  M_{\text{KV}}^{\text{orig}}
> >  = 2 \cdot 1024 \cdot 32 \cdot 1024 \cdot 2
> >  = 134{,}217{,}728 \text{ bytes}
> >  \approx 128 \text{ MiB},
> > $$
> > $$
> >  M_{\text{KV}}^{\text{ASTRA}}
> >  = 2 \Big(
> >  \frac{1024}{4} \cdot 32 \cdot 1024 \cdot 2
> > +
> > (4 - 1) \cdot \frac{1024}{4} \cdot 32 \cdot 32 \cdot \frac{10}{8}
> >  \Big)
> >  = 35{,}520{,}512 \text{ bytes}
> >  \approx 33.9 \text{ MiB}.
> >  $$
> >
> > Thus, in this configuration, ASTRA uses only about **26.5%** of the original KV cache memory. Detailed discussion  is included in Appendix G in the revised manuscript.
> >
> > **Q2. About Ring Attention.**
> >
> > **Astra is compatible with ring attention.** ASTRA builds on top ofsequence parallelism and transmits  vector-quantized KV indices instead of full-precision KV tensors. Ring attention follows  sequence parallelism, differing only in how KV tensors are passed around the ring and how attention computation is scheduled. Therefore, ASTRA can be naturally integrated with ring attention as a drop-in compression layer by communicating KV indices following the ring-style schedule.

---

### Author Response · Authors · 2025-12-03
**Summary to AC**

ASTRA is a communication-efficient framework for multi-device Transformer inference. It designs a Mixed-Precision Attention mechanism, computing local tokens at full precision while compressing remote tokens via vector quantization, to achieve up to $15.25\times$ speedup over existing methods in bandwidth-constrained environments (as low as 10 Mbps). Below we first summarize the core strengths of the work identified across reviews and then the concerns and how we address them.

**Addresses a Real and Important Problem.** Multiple reviewers (mHin, 24hX, 7h8e) acknowledged that the paper tackles a genuine bottleneck in multi-device Transformer inference for bandwidth-constrained environments. Reviewer mHin specifically noted that "communication dominates latency (58.6-93.5%) in bandwidth-constrained multi-device inference," making this a relevant problem for real-time AI applications.

**Novel Technical Approach.** Reviewers appreciated the Mixed-Precision Attention mechanism as a creative solution. Reviewer 7h8e found "the use of codebooks interesting," while Reviewer mHin highlighted the novelty of "using full-precision for local tokens and VQ for remote tokens".

**Strong Evaluation**: Reviewer mHin appreciated the "extensive experiments across multiple architectures (ViT, GPT-2), tasks (classification, language modeling), and conditions (bandwidth, device count, heterogeneity)" that demonstrate broad applicability.

---

We further provide a consolidated summary of our responses to all reviewers, focusing on clarifying concerns and resolving misunderstandings.

**Scalability to large LLMs (Llama-3-8B, Section 4.5).**
Following the feedback from reviewers 7h8e, mHin, 3hGB, we added comprehensive experiments to evaluate ASTRA with Llama3-8B. The results show that ASTRA scales to large models while maintaining task performance close to the original and delivering substantial speedups under low bandwidth.

**Robustness under non-ideal networks (Section 4.5).**
As suggested by reviewer mHin and 24hX, we further evaluated Llama-3-8B under (i) 5% random packet loss without retransmission and (ii) a 600-second variable-bandwidth trace (20–100 Mbps) generated from Pensieve. Results show that ASTRA maintains task accuracy under modest packet loss and delivers higher throughput than single-device and multi-device baselines when bandwidth fluctuates.

**Evaluation on downstream tasks (Appendix D).**
Following the feedback from reviewer 3hGB, we further conducted additional experiments on the downstream tasks to compare task performance between the original Llama3-8B and its ASTRA versions. The results demonstrate that our small increases in pre-training perplexity lead to minor differences on downstream tasks, validating ASTRA’s capability in maintaining task performance.

**Addressed clarifications and explanations.**
\- Motivation for edge deployment and limited bandwidth scenarios (Rev: 7h8e)
\- Clarification for task performance in zero-shot settings (Rev: mHin, Appendix H)
\- Baseline choice and fairness of comparisons (Rev: mHin)
\- Hyperparameter guidelines for practitioners (Rev: mHin, Appendix F)
\- Novelty clarification beyond “integration of known techniques” (Rev: 24hX)
\- Theoretical error bounds for attention under quantization and noise (Rev: 24hX, Appendix B and C)
\- Detailed memory cost analysis and ring attention compatibility explanation (Rev: 24hX, Appendix G)

We hope these additions and clarifications address the reviewers’ concerns and demonstrate that ASTRA is a practical, scalable, and well-founded framework for communication-efficient multi-device Transformer inference.

---

### Meta-Review · Area_Chair_mwZk · 2026-01-08

**Summary:**

Reviewers are exactly split, with supporters noting the important problem and creative approach, while critics highlight that experiments use only tiny outdated models (ViT, GPT-2) with simulation-only results. Authors addressed scale concerns in rebuttal by adding LLaMA-3-8B experiments showing the method scales, though one reviewer explicitly maintained their score. Despite improved evidence, the core concern about simulation-only evaluation without real multi-device deployment remains. I recommend rejection.

**Reviewer Concerns:**

see above

**Reviewer Scores:**

Discussion was sufficient; authors addressed scale concerns with new experiments but reviewers did not change scores, and scores would have remained similar.

---

### Decision · Program_Chairs · 2026-01-26

Reject